# Sociosexual behavior requires both activating and repressive roles of *Tfap2e*/AP-2ε in vomeronasal sensory neurons

**Jennifer M Lin**[1,2,†], **Tyler A Mitchell**[1,2,†], **Megan Rothstein**[3], **Alison Pehl**[1,2], **Ed Zandro M Taroc**[1,2], **Raghu R Katreddi**[1,2], **Katherine E Parra**[4], **Damian G Zuloaga**[4], **Marcos Simoes-Costa**[3], **Paolo Emanuele Forni**[1,2]*

[1]Department of Biological Sciences, University at Albany, State University of New York, Albany, United States; [2]The RNA Institute, University at Albany, Albany, United States; [3]Department of Molecular Biology and Genetics, Cornell University, Ithaca, United States; [4]Department of Psychology, University at Albany, State University of New York, Albany, United States

**\*For correspondence:**
pforni@albany.edu

[†]These authors contributed equally to this work

**Competing interest:** The authors declare that no competing interests exist.

**Abstract** Neuronal identity dictates the position in an epithelium, and the ability to detect, process, and transmit specific signals to specified targets. Transcription factors (TFs) determine cellular identity via direct modulation of genetic transcription and recruiting chromatin modifiers. However, our understanding of the mechanisms that define neuronal identity and their magnitude remain a critical barrier to elucidate the etiology of congenital and neurodegenerative disorders. The rodent vomeronasal organ provides a unique system to examine in detail the molecular mechanisms underlying the differentiation and maturation of chemosensory neurons. Here, we demonstrated that the identity of postmitotic/maturing vomeronasal sensory neurons (VSNs), and vomeronasal-dependent behaviors can be reprogrammed through the rescue of *Tfap2e*/AP-2ε expression in the *Tfap2e*[Null] mice, and partially reprogrammed by inducing ectopic *Tfap2e* expression in mature apical VSNs. We suggest that the TF *Tfap2e* can reprogram VSNs bypassing cellular plasticity restrictions, and that it directly controls the expression of batteries of vomeronasal genes.

## Editor's evaluation

Lin et al. investigate the role of AP-2e in vomeronasal sensory neurons through targeted gene deletion and rescue. They report that knockout of AP-2e reduces expression of basal markers, while induced expression can rescue basal identity. Moreover, forced expression of AP-2e in mature apical neurons causes them to express some basal markers. Future studies are needed to understand the impact of mutations on VNO receptor expression, and mechanistically why behavioral changes are observed.

## Introduction

Neuronal differentiation is controlled by the selective expression of transcription factors (TFs), chromatin modifiers, and other regulatory factors that reduce cellular plasticity. During neuronal differentiation, terminal selectors can activate identity-specific genes that define functional properties specific to a particular neuronal type. However, reprograming postmitotic neurons by ectopically expressing terminal selectors in *Caenorhabditis elegans* suggests that the 'reprogrammability' of neurons is

progressively lost during postembryonic life (*Patel and Hobert, 2017*; *Patel et al., 2012*; *Rahe and Hobert, 2019*). This reduction in cellular plasticity may arise from chromatin modifications that prevent the activation of alternative differentiation programs. However, sensory neurons of the vomeronasal organ (VNO) in rodents can undergo some postnatal reprogramming following the aberrant expression of TFs (*Lin et al., 2018*).

The accessory olfactory system (AOS) contains the VNO, which is primarily responsible for detecting odors and chemosignals that trigger social and sexual behaviors (*Trouillet et al., 2019*; *Trouillet et al., 2021*). The vomeronasal sensory epithelium of rodents is mainly composed of VSNs. The VSN populations selectively express only one or two receptors encoded by the two vomeronasal receptor (VR) gene families: V1R and V2R (*Dulac and Axel, 1995*; *Herrada and Dulac, 1997*; *Matsunami and Buck, 1997*; *Ryba and Tirindelli, 1997*). V1R- and V2R-expressing neuronal populations each detect distinct chemosignals, induce different innate behaviors, show distinct localization patterns in the VNO, and project to specific areas of the accessory olfactory bulb (AOB) (*Cloutier et al., 2002*; *Dulac and Axel, 1995*; *Isogai et al., 2011*; *Katreddi and Forni, 2021*; *Mohrhardt et al., 2018*; *Mombaerts et al., 1996*; *Stowers et al., 2002*). The V2R-expressing neurons localize to the basal portions of the vomeronasal epithelium (VNE) and around the vasculature (*Naik et al., 2020*), while V1R-expressing neurons localize to the apical part. Basal and apical VSNs continually regenerate from common pools of Achaete Scute like-1 (Ascl1)-positive neural progenitor cells localized in the lateral and basal margins of the VNE (*Cau et al., 1997*; *de la Rosa-Prieto et al., 2010*; *Katreddi and Forni, 2021*; *Martínez-Marcos et al., 2000*; *Murray et al., 2003*). However, we are only starting to understand how the apical and basal VSN cell differentiation programs are initiated and which factors aide in maintaining apical and basal neuronal identity (*Enomoto et al., 2011*; *Katreddi et al., 2022*; *Lin et al., 2018*; *Naik et al., 2020*; *Oboti et al., 2015*).

Establishing functional basal and apical VSNs is crucial for intra- and interspecies social interactions in rodents. Deficits in basal neuron functionality prevented sex discrimination, reduced male-male and maternal aggressive behaviors, and inhibited the detection of predator odors (*Chamero et al., 2011*; *Stowers et al., 2002*).

TFs can drive cellular processes that control the expression of genes defining their cellular and functional identity. The AP-2 family of TFs is comprised of five members: AP-2α, AP-2β, AP-2γ, AP-2δ, and AP-2ε, which are encoded by distinct genes (*Tfap2a, Tfap2b, Tfap2c, Tfap2d, Tfap2e*) (*Eckert et al., 2005*; *Pellikainen and Kosma, 2007*; *Wankhade et al., 2000*). AP-2 family members play critical roles during development, such as contributing to neural crest differentiation (*Luo et al., 2020*; *Rothstein and Simoes-Costa, 2020*), cell specification, limb development, and organogenesis (*Bassett et al., 2012*; *Chambers et al., 2019*; *Kantarci et al., 2015*). Some AP-2 family members may have pioneer factor properties (*Fernandez Garcia et al., 2019*; *Rothstein and Simoes-Costa, 2020*; *Seberg et al., 2017*; *Williams et al., 2009*).

Aside from *Tfap2e*, Notch signaling and Bcl11b control Gαo+VSNs' differentiation, homeostasis, and survival (*Enomoto et al., 2011*; *Katreddi et al., 2022*). We previously proposed that *Tfap2e*, which is only expressed after the apical and basal VSN dichotomy is established, is necessary for further specification of basal VSN identity (*Lin et al., 2018*). Using mice expressing non-functional AP-2ε, we discovered that VSNs can still acquire the Gαo+/basal identity; however, these VSNs have reduced survival and can acquire some Gαi2+/apical VSNs' molecular features over time (*Lin et al., 2018*). While we examined the role of *Tfap2e* in maintaining cellular identity and homeostasis of the VNE, critical outstanding questions remain unresolved. What role does this TF actively play to control the basal genetic program? What is the extent of cellular plasticity in differentiated neurons in mammals (*Patel and Hobert, 2017*; *Rahe and Hobert, 2019*)?

Here, we aimed to understand (1) if AP-2ε functions as a terminal selector factor for basal VSNs, (2) how much cellular plasticity postmitotic neurons retain once differentiated, and (3) to what extent genetic dysregulation in mature VSNs translates into behavioral changes. We generated a Cre-inducible mouse line, where we inserted the *Tfap2e* gene into the *Rosa26* locus. Using this knock-in mouse line, we could (1) rescue the *Tfap2e* knockout (KO's) VNO morphology and functionality and (2) ectopically express *Tfap2e* in maturing Gαi2+/apical VSNs. This approach enabled us to assess its ability to reprogram differentiated apical VSNs to basal VSNs. By combining histological analyses, behavioral assessments, and single-cell RNA sequencing (scRNA-seq) analysis, we examined whether *Tfap2e* functions as a master regulator to reprogram differentiated neurons and alter animal behaviors.

In addition, we used CUT&RUN (*Skene et al., 2018*) to identify direct genetic targets of AP-2ε that controls the basal VSNs' identity program. Overall, we suggest that AP-2ε partially functions as a terminal selector by activating some basally enriched genes while simultaneously suppressing specific apically enriched genes.

## Results

### Transcriptome differences between apical and basal VSNs

Using scRNA-seq on VNOs from *Omp^Cre* heterozygous control mice at P10, we identified key features of VSNs based on the expression plots. We then clustered single cells into representative Uniform Manifold Approximation and Projections (UMAPs) (*Figure 1*). *Ascl1* (*Figure 1A*), *Neurogenin1* (*Neurog1*) (*Figure 1B*), and *Neurod1* (*Figure 1C*) expression identified proliferative VSN progenitors and precursors (*Katreddi and Forni, 2021*). We determined that the dichotomy of apical-basal differentiation begins when the cells transition from *Neurog1* to *Neurod1* expression, and during the *Neurod1* phase (*Figure 1B and C*; *Katreddi et al., 2022*). The later stages of apical and basal VSN maturation were marked by the expression of *Gap43* (*Figure 1D*) in immature VSNs and *Omp* (*Figure 1E*) in more mature VSNs (*Katreddi and Forni, 2021*).

Consistent with prior reports, the mRNA for the TF *Bcl11b* (*Figure 1F*) was found to be expressed in *Neurog1+/Neurod1+* (*Figure 1B and C*) precursors and in differentiating apical and basal VSNs. While *Bcl11b* was expressed as a continuum along the basal differentiation trajectory, in the apical neurons, *Bcl11b* was not expressed until later stages of maturation (*Figure 1F*; *Enomoto et al., 2011*; *Katreddi et al., 2022*). Interestingly, the apical VSN-specific TF *Meis2* was expressed in progenitor cells along with apical VSNs' differentiating neurons (*Figure 1G*). In addition, apical VSNs also express *Gnai2*/Gαi2, *Nrp2*, and V1Rs at more mature stages (*Figure 1I–K*). On the other side, mRNA for *Tfap2e* (*Figure 1K*) was found to be expressed, in line with our prior works (*Katreddi et al., 2022*; *Lin et al., 2018*), in maturing and mature basal VSNs. The basal VSNs' differentiation trajectory was further confirmed by the expression of known V2R/basal VSNs' markers such as *Robo2*, *Gnao1*/Gαo, and *Vmn2r* receptors (*Figure 1K–N*). In addition, our transcriptome analysis revealed significant enrichments (q<0.05, *Figure 1Q*, *Supplementary file 1*) of several previously unreported genes in either apical or basal VSNs. *Fbxo17*, *Mt3* (*Figure 1O and P*; *Figure 4—figure supplement 3*), and *Keratin18* (*Krt18*) were among the genes that we found enriched in maturing basal VSNs.

### Inducible R26AP-2ε rescues basal VSNs in AP-2ε KOs

We hypothesized that Tfap2e can control gene expression during basal VSN maturation. So, we generated a new Cre-inducible mouse line (B6.Cg-Gt(ROSA)26Sor^tm(CAG-mTfap2e)For). We inserted a loxP-flanked stop cassette to prevent the transcription of a CAG promoter-driven murine *Tfap2e* gene, which was knocked into the first intron of the Gt(ROSA)26Sor locus (*Figure 2*). We refer to this as R26AP-2ε. AP-2ε expression is normally restricted to basal regions of the VNO with higher expression levels of AP-2ε in the neurogenic marginal zones (*Enomoto et al., 2011*; *Lin et al., 2018*; *Naik et al., 2020*; *Figures 1 and 2B,B′*). To test our Cre-inducible AP-2ε line, we performed anti-AP-2ε immunostaining on wild-type (WT) controls, *Tfap2e^Cre/Cre^* (*Tfap2e^Null^*) (*Feng et al., 2009*; *Lin et al., 2018*), and *Tfap2e^Cre/Cre^/R26AP-2ε* (*Tfap2e^Rescue^*) mice (*Figure 2B–D*). As expected (*Lin et al., 2018*), WT mice showed AP-2ε immunoreactivity in the basal regions of the VNE with strong immunoreactivity in the neurogenic regions (*Figure 2B and B′*). However, in *Tfap2e^Null^* mice, where Cre was knocked into the DNA binding domain of *Tfap2e* (*Feng et al., 2009*; *Feng and Williams, 2003*), we observed faint AP-2ε cytoplasmic immunoreactivity limited to the most marginal zones of the VNO and no immunoreactivity in the rest of the neuroepithelium (*Figure 2C and C′*). *Tfap2e^Rescue^* mice showed restored AP-2ε immunoreactivity in the basal region of the VNE (*Figure 2D*). However, we observed that the AP-2ε expression pattern and immunoreactivity were not identical to controls in the neurogenic regions. In *Tfap2e^Rescue^* mice, we observed no AP-2ε immunoreactivity at the tips of the neurogenic niche in the VNE (*Figure 2D′*), suggesting a delayed AP-2ε expression after *Tfap2e^Cre^*-mediated recombination compared to controls (*Figure 2B′ and D′*).

By analyzing the expression of the basal markers downregulated in *Tfap2e* KOs (*Lin et al., 2018*), such as V2R2 (*Figure 2E–G*) and Gαo (*Figure 2H–J*), we confirmed restored expression in the rescued KOs. Cell quantifications indicated a significant increase in the number of basal cells expressing

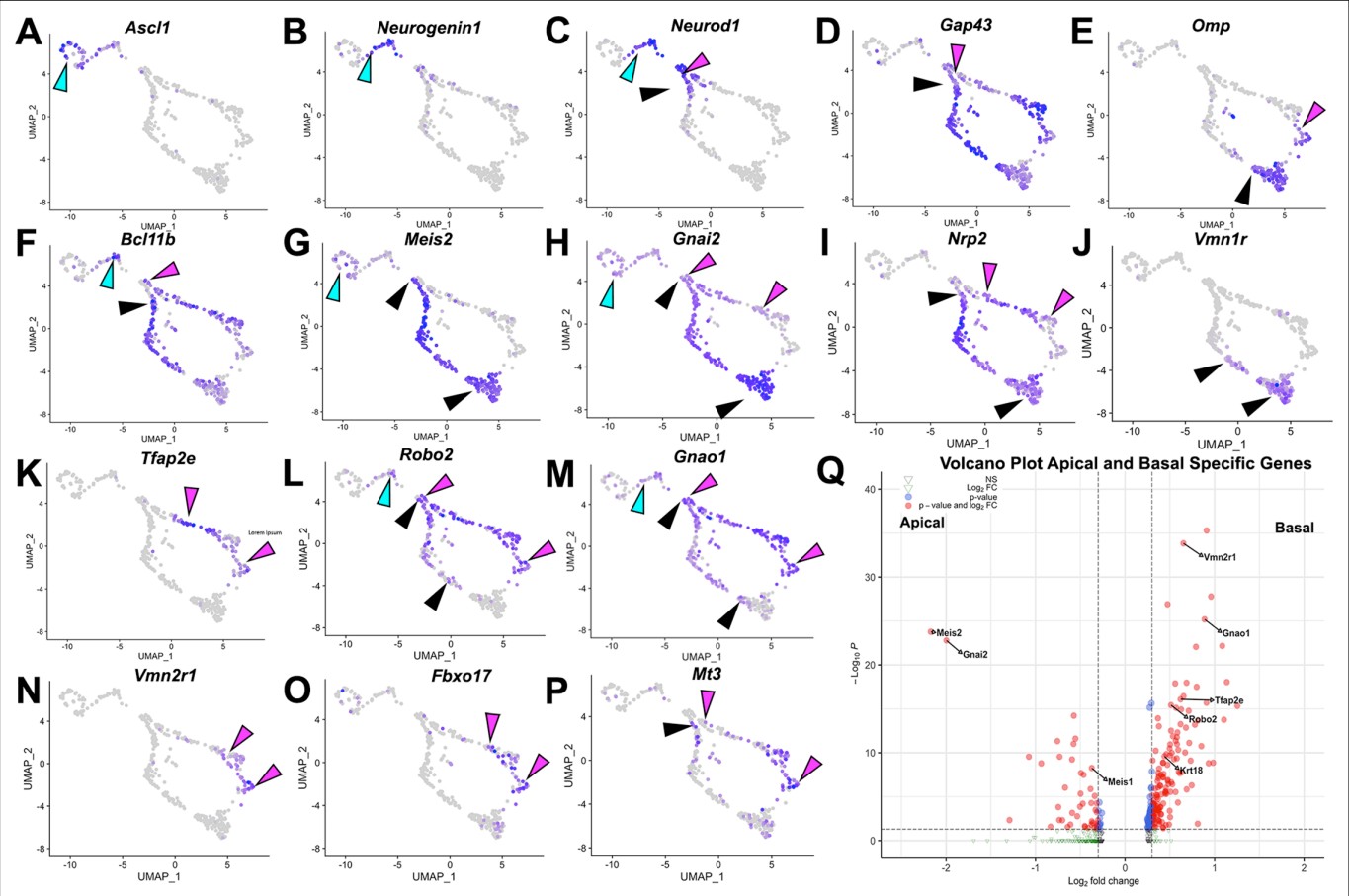

**Figure 1.** Analysis of single-cell sequencing data of the vomeronasal organ. P10 male controls shows the developing vomeronasal neurons as they progress through (**A–C**) neurogenesis (*Ascl1, Neurog1, Neurod1*), (**D–F**) maturation (*Gap43, Omp, Bcl11b*), (**G–J**) apical vomeronasal sensory neurons (VSNs)' differentiation/maturation (*Meis2, Gnai2, Nrp2, Vmn1r*) and (**K–P**) basal VSNs' differentiation/maturation (*Tfap2e, Robo2, Gnao1, Vmn2r1, Fbxo17, Mt3*). (**A**) *Ascl1* is expressed by transiently amplifying progenitor cells (cyan arrow), which transition into the immediate neuronal precursors that turn on pro-neural genes (cyan arrow) *Neurog1* (**B**) and *Neurod1* (**C**) and turn off as the precursors turn into immature neurons (black and magenta). (**D**) Immature neurons express *Gap43*, which persists in both apical (black arrow) and basal (magenta arrow) branches, until it declines as neurons begin to reach maturity and express (**E**) *Omp* in mature apical (black arrow) and mature basal (magenta arrow) VSNs. *Gap43* and *Omp* expression briefly overlap, as the neurons transition to a fully differentiated mature stage. (**F**) *Bcl11b* mRNA expression is found in both apical and basal VSNs but at different developmental timepoints. *Bcl11b* is found in committed basal precursors near the establishment of the apical/basal dichotomy (magenta arrow) but is not found until later in apical VSN development (black arrow). (**G**) *Meis2* mRNA expression is found in the apical branch (black arrows) and even in early neurogenesis stages (cyan arrow), and their expression does not overlap. (**H**) *Gnai2* expression starts in immature apical and basal VSNs. However, its expression increases in mature apical VSNs but fades in mature basal VSNs. (**I**) *Nrp2* expression is only retained in mature apical VSNs. (**J**) *Vmn1r* is only expressed in maturing/mature apical VSNs. (**K**) *Tfap2e* is expressed by the basal branch (magenta arrows). (**L**) *Robo2* expression is only retained in mature basal VSNs. (**M**) *Gnao1* expression starts in immature apical and basal VSNs. However, its expression increases in mature basal VSNs but fades in mature apical VSNs. (**N**) *Vmn2r1* is restricted to maturing/mature basal VSNs. (**O**) *Fbxo17* is mainly expressed in mature/maturing basal VSNs. (**P**) *Mt3* is mainly expressed in mature/maturing basal VSNs. (**Q**) Enhanced volcano plot. Differential gene expression between apical and basal branches of VSNs. Apical-specific genes (55 genes) trend left and basal-specific genes (187 genes) trend right. Significance defined as log2-fold change >0.3 and adjusted p-value ≤0.05.

basal markers in *Tfap2e^Rescue* compared to *Tfap2e^Null* mice, though the number of basal VSNs in the *Tfap2e^Rescue* was smaller when compared to controls (**Figure 2N**). *Krt18* is normally enriched in basal neurons (**Figure 1Q**). Immunohistochemistry confirmed Krt18 protein expression in the basal territories of the VNO (**Figure 2K**). We observed reduced Krt18 immunoreactivity in *Tfap2e^Null* mice and restored expression in *Tfap2e^Rescue* mice (**Figure 2L–N**). However, in *Tfap2e^Rescue* mice, Krt18 still showed lower expression levels than in WT mice (**Figure 2N**). Taken together, we conclude that exogenous AP-2ε in postmitotic VSNs can partially rescue the expression of basal VSN markers in *Tfap2e^Null* mice. Rescue

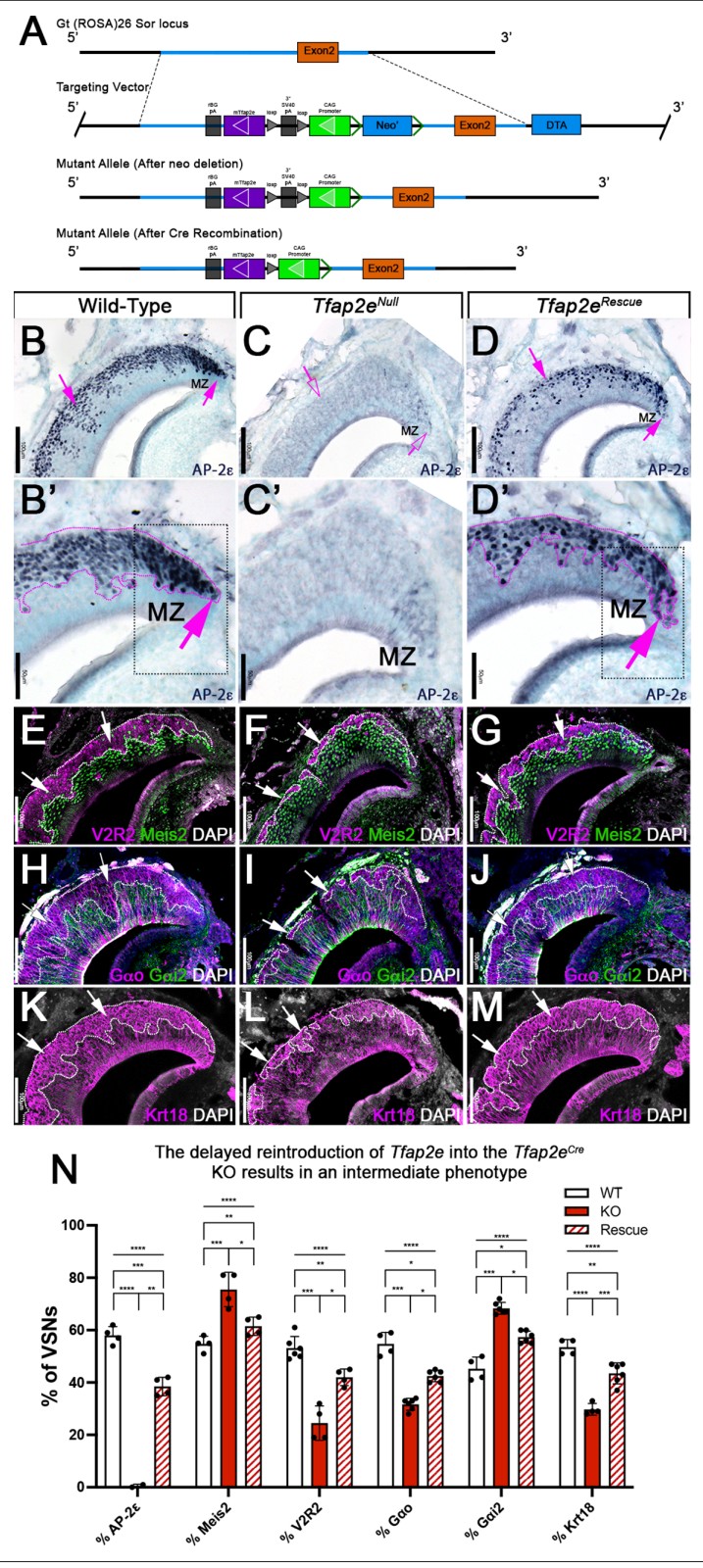

**Figure 2.** R26AP-2ε mouse line mouse generation and characterization of rescued *Tfap2e* expression in *Tfap2e^Null^* mice. (**A**) Knock-in strategy through homologous recombination to generate the R26AP-2ε mouse line. The *CAG-loxP-stop-loxP-mouse-Tfap2e* cassette as integrated into the first intron of *Rosa26*. (**B–D'**) Immunohistochemistry on P21 wild-type (WT) (**B, B'**), *Tfap2e^Null^* (**C,C'**), and *Tfap2e^Rescue^* (**D,D'**) mice against AP-2ε. (**B,B'**) In WT mice AP-2ε

*Figure 2 continued on next page*

*Figure 2 continued*

expression is in the marginal zones (MZ) and in the basal regions of the VSNs. (**C,C'**) In *Tfap2e^Null^* mice, some AP-2ε immunoreactivity is observed in the MZ but lost in the central regions where more mature neurons reside. (**D,D'**) In the *Tfap2e^Rescue^*, AP-2ε is expressed in the basal region, but with less intensity and density at the MZ and in central regions compared to WT controls. (**E–G**) Immunostainings against V2R2 (magenta) and Meis2 (green) counterstained with 4',6'-diamidino-2-phenylindole (DAPI) (white). (**H–J**) Immunostainings against Gαo (magenta) and Gαi2 (green) counterstained with DAPI. (**K–M**) Immunostainings against Krt18 (magenta) counterstained with DAPI (white). (**N**) Quantifications of the percentage of vomeronasal sensory neurons (VSNs) expressing apical/basal markers in WT, *Tfap2e^Null^*, and *Tfap2e^Rescue^* mice. *Tfap2e^Null^* mice show a dramatic reduction in basal VSNs and apical VSNs occupy most of the epithelium. The *Tfap2e^Rescue^* has an intermediate phenotype between WT and *Tfap2e^Null^* mice, where the vomeronasal epithelium (VNE) contains more basal VSNs than in the *Tfap2e^Null^* mice but does not reach the equivalency of the WT ($p<0.05$ = *, $p<0.01$ = **, $p<0.001$ = ***, $p<0.0001$ = ****). One-way ANOVA. Error bars are standard deviation. N=4 for WT in % AP-2ε, % meis2, % Gαo, % Gαi2, and % Krt18. N=6 for WT in % V2R2. N=2 for *Tfap2e^Null^* in % AP-2ε. N=4 for *Tfap2e^Null^* in % Meis2, % V2R2, and % Krt18. N=6 for *Tfap2e^Null^* in % Gαo and % Gαi2. N=4 for *Tfap2e^Rescue^* in % AP-2ε, % Meis2, and % V2R2. N=6 for *Tfap2e^Rescue^* in % Gαo, % Gαi2, and % Krt18.

of the *Tfap2e^null^* phenotype indicates that our inducible R26AP-2ε mouse line is a suitable model for conditional expression of functional AP-2ε.

## Re-expressing AP-2ε in *Tfap2e^Null^* mice rescue social behaviors

The specification and organization of VSNs and their respective circuit assembly in the AOB are essential to trigger a variety of social and sexual behaviors (*Chamero et al., 2011*; *Chamero et al., 2007*; *Stowers et al., 2002*; *Trouillet et al., 2019*). We speculated that *Tfap2e^Null^* mice could not discriminate between urine of different sexes. Thus, we performed an odorant preference test. In this test, individual mice were simultaneously presented with male and female whole urine for a 2 min period (*Figure 3*). WT male mice showed a significant preference for urine from the opposite sex (*Figure 3B*; *Pankevich et al., 2004*; *Stowers et al., 2002*). However, *Tfap2e^Null^* male mice did not display significant preference for female urine, confirming a loss of function (LOF) of basal VSNs (*Lin et al., 2018*) and consequently a reduced ability to discriminate between urine from either sex (*Figure 3B*; *Pankevich et al., 2004*; *Stowers et al., 2002*). In line with our histological results (*Figure 2D, G, J and M*), *Tfap2e^Rescue^* male mice showed a significant preference for female urine similar to WT controls (*Figure 3B*).

To further investigate the behavioral outcome of *Tfap2e* LOF, we performed a resident intruder assay for intermale aggression (*Figure 3C*; *Chamero et al., 2011*; *Montani et al., 2013*; *Stowers et al., 2002*). *Tfap2e^Null^* mice displayed significantly reduced aggressive behavior compared to WT controls (WT [mean # of attacks = 9.5 SE±0.7] vs. *Tfap2e^Null^* [mean # of attacks = 1.1 SE±0.2] t-test p=0.0138). Yet, when male *Tfap2e^Rescue^* mice were exposed to male intruders, they displayed restored aggressive behavior (*Tfap2e^Null^* [mean # of attacks = 1.1 SE±0.2] vs. *Tfap2e^Rescue^* [mean # of attacks = 22.8 SE±3.4] t-test p=0.0168). ANOVA test with post hoc analysis confirmed that *Tfap2e^Rescue^* mice significantly differed in their aggressive behavior from *Tfap2e^Null^* (p=0.0252) but not from controls (*Figure 3D*).

We measured the mass of the seminal vesicles from each genotype to rule out any changes in general androgen levels, which may explain any potential behavioral differences (*Zuloaga et al., 2007*). We found no significant differences when the seminal vesicle weights were normalized to the total body weight of each mouse (*Figure 3E*). Taken together, these data suggest that re-expression of *Tfap2e* in KO mice can reestablish the functional properties of basal VSNs.

## Ectopic expression of *Tfap2e* in mature apical VSNs increases the expression of basal enriched genes

Several AP-2 family members have been proposed to have pioneer activity (*Fernandez Garcia et al., 2019*; *Rothstein and Simoes-Costa, 2020*; *Seberg et al., 2017*; *Williams et al., 2009*). We tested whether ectopic *Tfap2e* expression can alter the transcriptomic profile of maturing neurons. *Olfactory marker protein* (*Omp*) is an accepted marker for postmitotic/maturing olfactory and VSNs (*Buiakova et al., 1994*; *Enomoto et al., 2011*; *Farbman and Margolis, 1980*). By analyzing our scRNA-seq data from *Omp^Cre^* mice, we confirmed that *Omp* mRNA expression can be detected in maturing apical neurons shortly after the apical basal dichotomy is established (*Figure 1E*). Thus, we used an *Omp^Cre^*

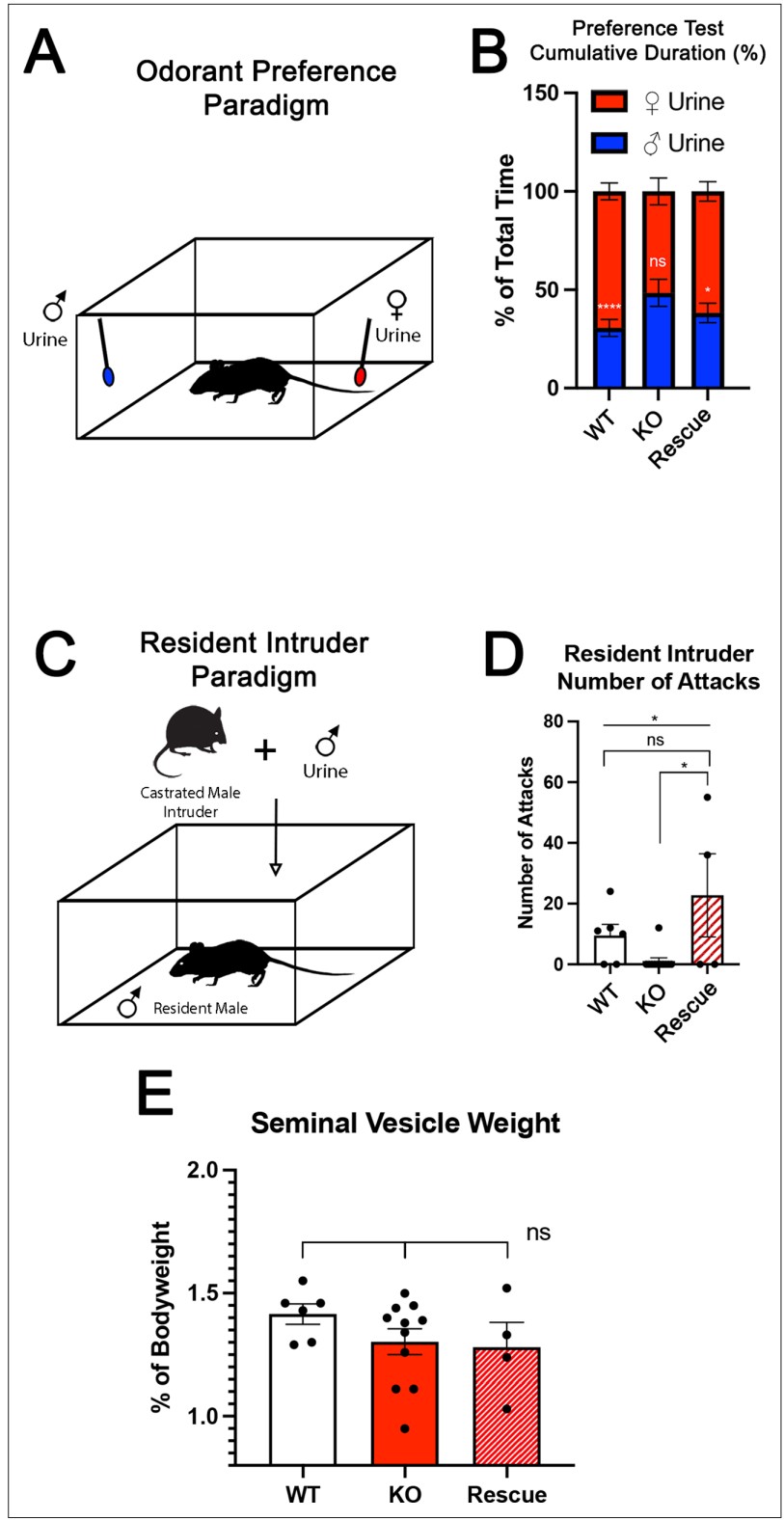

**Figure 3.** Territorial aggression and sex preference depends on *Tfap2e* expression in mice. (**A**) The odorant preference paradigm where cotton swabs with either male or female whole urine are placed on opposite ends of a test cage and the amount of time spent smelling each odorant is measured. (**B**) Male wild-type (WT) mice spent significantly more time investigating female odorants than male odorants. Preference for female odorants is lost in *Tfap2e* knockout (KO) male mice but restored in *Tfap2e*^*Rescue*^ mice (p<0.05 = *, p<0.01 = **, p<0.001 =

*Figure 3 continued on next page*

*Figure 3 continued*

\*\*\*). Unpaired t-test. Error bars are standard deviation. N=6 for WT, n=11 for *Tfap2e^Null^* and n=4 for *Tfap2e^Rescue^*. (**C**) Male-male aggression was evaluated using the resident intruder paradigm. (**D**) WT mice display aggressive behaviors toward male intruders and number of attacks were quantified. *Tfap2e^Null^* mice attacked intruders significantly less than WT mice. However, *Tfap2e^Rescue^* mice showed significantly more aggressive behaviors that is not significantly different than the WT male mice (p<0.05 = \*). One-way ANOVA. Error bars are standard deviation. N=6 for WT, n=11 for KO, n=4 for rescue. (**E**) Seminal vesicle weight was not significantly different across all genotypes when normalized to body weight. One-way ANOVA. Error bars are standard deviation. N=6 for WT, n=11 for KO, n=4 for rescue.

mouse line to drive expression of *Tfap2e* (*Omp^Cre^*/R26AP-2ε) in all olfactory and vomeronasal neurons (**Figure 4A and D**). In this manuscript, we will often refer to *Omp^Cre^*/R26AP-2ε as ectopic mutants. Notably, both apical and basal VSNs express several known *Tfap2* cofactors, including *Cited2* and *Ep300* (**Bamforth et al., 2001**; **Bragança et al., 2003**; **Eckert et al., 2005**), suggesting that both VSN populations are molecularly competent for functional AP-2ε transcriptional activity (**Figure 4—figure supplement 1**).

Immunostaining against AP-2ε showed no immunoreactivity in the main olfactory epithelium (OE) of control animals (**Figure 4—figure supplement 2**). However, in *Omp^Cre^*/R26AP-2ε mutants, the OE expressed immunodetectable AP-2ε (**Figure 4—figure supplement 2**). *Omp^Cre^* controls and *Omp^Cre^*/R26AP-2ε mutants displayed comparable gross morphology of the OE with no ectopic V2R immunoreactivity (**Figure 4—figure supplement 2**).

In the VNO of *Omp^Cre^* controls, AP-2ε was found only in cells in the basal territory (**Figure 4A, A1**). However, in *Omp^Cre^*/R26AP-2ε mice, we found that virtually all the VSNs expressed AP-2ε (**Figure 4D and D1**). In these mutants we also observed AP-2ε expression in sparse sustentacular cells lining the lumen of the VNO (**Figure 4D**).

When comparing the VNO of P10 controls (either WT or *Omp^Cre^*) and *Omp^Cre^*/R26AP-2ε mutants, we observed that the ectopic mutants had a significantly broader *Gnao1* mRNA expression across the VNE (**Figure 4B and E**). As the broader expression was hardly conducible to individual cells, we performed a densitometric analysis of sections after in situ hybridization (ISH). This indicated that, in the mutants, a larger percentage of the VNE was positive for *Gnao1* expression (**Figure 4G**). Moreover, we found that *Omp^Cre^*/R26AP-2ε mutants had a larger number of cells immunodetectable using the anti-V2R2 antibody (**Silvotti et al., 2011**; **Figure 4C, F and H**). These immunoreactive cells could be found spanning from basal VNO regions to the lumen.

In line with these observations, scRNA-seq data from *Omp^Cre^* controls and *Omp^Cre^*/R26AP-2ε mutants indicated that apical VSNs expressing AP-2ε had variable, but significant (p<0.05) upregulation of *Gnao1* mRNA as well as increased expression for some C family *Vmn2r*, many of which can be detected using the anti-V2R2 antibody (**Figure 4I and J**; **Figure 4—figure supplement 2**; **Silvotti et al., 2011**). Notably our scRNA-seq data indicated that that low/basal levels of *Vmn2r7* mRNA expression can also be detected in sparse apical neurons of controls (**Figure 4I**; **Figure 4—figure supplement 2**).

Immunostaining against V2R2 and Meis2 also highlighted that, while in controls, Meis2 and V2R2 remained segregated (**Figure 4K**), in the ectopic mutants, V2R receptors could be immunodetected in Meis2+ apical neurons (**Figure 4L**). These data suggest that ectopic *Tfap2e* expression can induce or increase the expression of basal enriched genes.

In order to better follow the effects of ectopic *Tfap2e* expression on apical cells, we also performed RNAscope analysis using probes against the basal VSN marker *Gnao1* and the apical markers *Gnai2* and *Meis2* (**Figure 4M and N**).

In controls (**Figure 4M**), we could observe a clear segregation between the basal VSNs positive for *Gnao1* and the apical VSNs positive for *Meis2* and *Gnai2*. Notably *Meis2* was also expressed in the sustentacular cells and in newly formed cells in the marginal zone. However, in *Omp^Cre^*/R26AP-2ε mutants (**Figure 4N**), we could observe, as after regular ISH (**Figure 4E**) a low but obvious expansion of *Gnao1* expression to the apical domains of the VNO. Signal highlighting *Gnao1* expression could be found in *Meis2* and *Gnai2* positive cells.

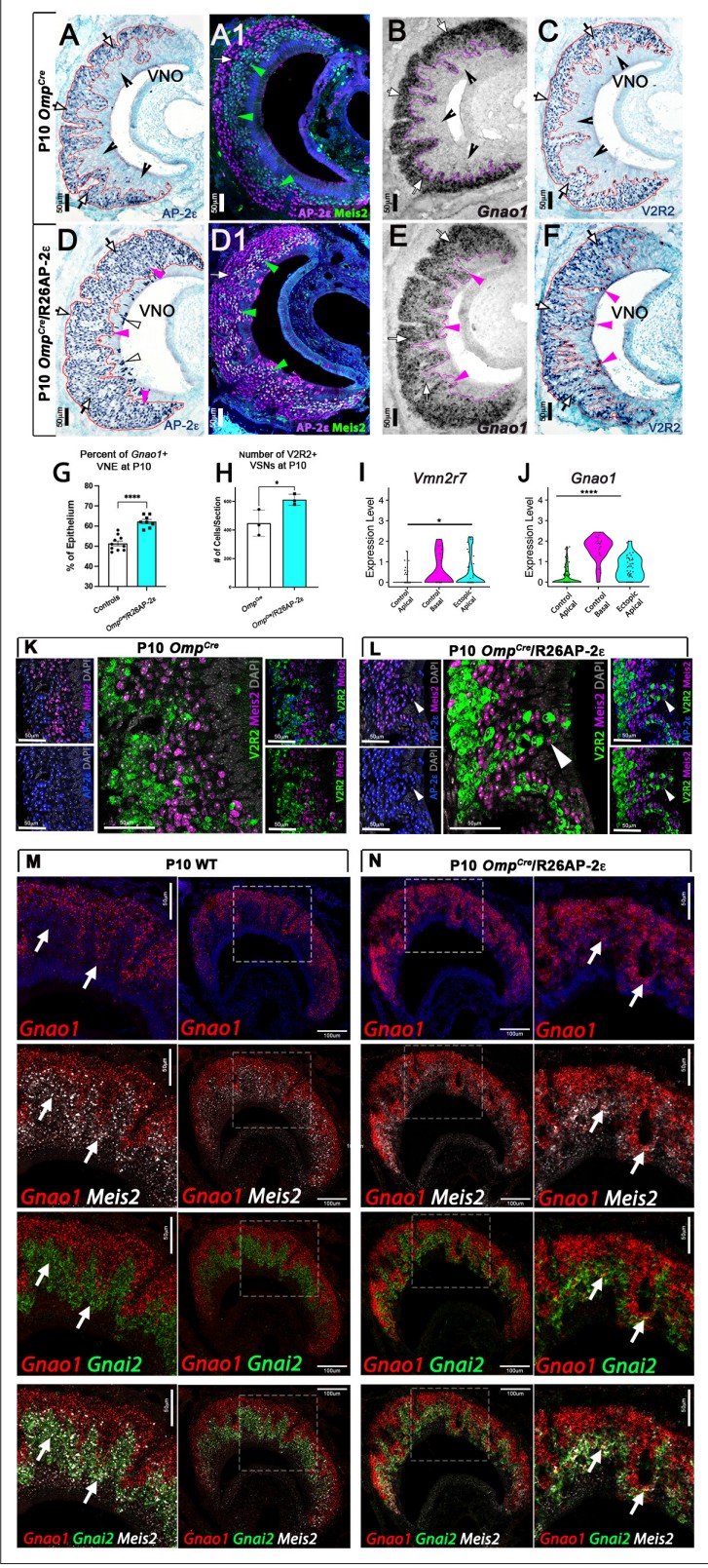

**Figure 4.** Ectopic expression of *Tfap2e* in the main olfactory epithelium (MOE) and the vomeronasal epithelium (VNE) promotes expression of basal markers. Immunostainings at P10 on *Omp^Cre* controls (**A–C**) and *Omp^Cre*/R26AP-2ε mutants (**D–F**). (**A**) IHC against AP-2ε in the vomeronasal organs (VNOs) shows that AP-2ε is expressed in only the basal vomeronasal sensory neurons (VSNs) (arrow) in controls but have extended AP-2ε

*Figure 4 continued on next page*

*Figure 4 continued*

immunoreactivity into apical (magenta arrowheads) and sustentacular regions (white arrowheads) in ectopic mutants (**D**). (**A1**) AP-2ε, Meis2 double immunofluorescence shows segregated AP-2ε (white arrow) and Meis2 (green arrowheads) in controls. (**D1**) In the ectopic mutants AP-2ε expression is detected in Meis2+ cells (green arrowheads). (**B,E**) ISH against *Gnao1* show that in controls (**B**) *Gnao1* mRNA expression is restricted to the basal regions of the VNE. (**E**) Ectopic mutants show *Gnao1* mRNA reactivity in the apical regions of the VNE (magenta arrowheads). (**C,F**) Immunohistochemistry against V2R2 in the VNO of controls (**C**) shows no immunoreactivity in the apical regions of the VNE in controls (notched arrows) and is limited to the basal VSNs (white arrows). (**F**) In mutants, more of the VNE was positive for V2R2 in mutants, as expression expands into the apical regions of the epithelium (magenta arrowheads). (**G**) Quantifications at P10 show a significant increase ($p<0.0001$ = ****) in the amount of the neuroepithelium positive for *Gnao1* in $Omp^{Cre}$/R26AP-2ε mutants. Significance calculated using arcsine transformation on percent values; unpaired two-tailed t-test analysis. Error bars are standard deviation. N=10 for $Omp^{Cre}$ controls and n=8 for $Omp^{Cre}$/R26AP-2ε mutants. (**H**) Quantifications at P10 show a significant increase in the number of the VSNs positive for V2R2 in $Omp^{Cre}$/R26AP-2ε mutants. Unpaired t-test. Error bars are standard deviation. N=3 for both $Omp^{Cre}$ controls and $Omp^{Cre}$/R26AP-2ε mutants ($p<0.05$ = *). (**I,J**) Violin plot of *Vmn2r7* (**I**), and *Gnao1* (**J**) mRNA expression between apical and basal VSNs in $Omp^{Cre}$ controls and apical VSNs of $Omp^{Cre}$/R26AP-2ε mutant mice show significant upregulation of these basal markers in mutants based on p-value ($p<0.05$ = *, $p<0.0004$ = ****). (**K,L**) Immunofluorescence against V2R2 (magenta) and Meis2 (green) in controls (**K**) and $Omp^{Cre}$/R26AP-2ε mutants (**L**). Arrows indicate a Meis2+ cell immunoreactive against anti-V2R2 antibodies in mutants (**L**). (**M,N**) Single-molecule FISH (RNAscope) against *Gnao1, Gnai2, and Meis2* of P10 wild-type (WT) and $Omp^{Cre}$/R26AP-2ε mutants. In WTs (**M**), a clear segregation between the *Gnao1*+ (red) basal VSNs and the apical cells positive for *Meis2*+ (white) and *Gnai2*+ (green) could be seen. Arrows show *Meis2*+/*Gnai2*+ apical VSNs in control VNO tissue negative for *Gnao1*. In $Omp^{Cre}$/R26AP-2ε mutants (**N**) low but obvious expansion of *Gnao1* (red) expression to the apical domains of the VNO. Signal highlighting *Gnao1* expression could be found in *Meis2*+ (white) and *Gnai2*+ (green) cells. The arrows point to *Meis2*+/*Gnai2*+ apical VSNs with ectopic expression of *Gnao1*.

The online version of this article includes the following figure supplement(s) for figure 4:

**Figure supplement 1.** Expression of potential AP-2 cofactors in both apical and basal vomeronasal sensory neurons (VSNs).

**Figure supplement 2.** AP-2ε and V2R2 immunoreactivity in the MOE (main olfactory epithelium).

**Figure supplement 3.** Ectopic expression in apical vomeronasal sensory neurons (VSNs) of indicated genes in $Omp^{Cre}$/R26AP-2ε mutants.

## Ectopic *Tfap2e* expression leads to a progressive disorganization of the VNE

In ectopic *Tfap2e* mutants, at P21 and more dramatically at 3 months of age, we noticed an increasing level of cellular disorganization of the VNE that is not seen in controls (*Figure 5*) with: (1) VSNs spanning from the basal territories to regions of the lumen devoid of Sox2+ sustentacular cells, and (2) ectopic sustentacular cells organized in spherical structures or intraepithelial cysts with a subsidiary lumen within apical and basal territories (*Figure 5B–C'*). Notably, the regions with ectopic sustentacular cells appeared to be mostly surrounded by apical VSNs expressing AP-2ε, Meis2, and Sox2 and were enriched in the intermediate zones of the VNE (*Figure 5B–D*). Interestingly, a low level of Sox2 immunoreactivity was observed in apical VSNs in both controls and $Omp^{Cre}$/R26AP-2ε mice with higher intensity in cells closer to the sustentacular cell layer (*Figure 5—figure supplement 1*).

The affinity and positioning of epithelial cells are largely dictated by the expression of surface adhesion molecules (*Fagotto, 2014*; *Polanco et al., 2021*). Transcriptome comparison of $Omp^{Cre}$/R26AP-2ε mutants and controls suggest that the aberrant cell positioning in the VNE of mutants can arise from broad variations in expression levels of multiple adhesion molecules throughout *Meis2*+ cells (*Figure 5E*).

Furthermore, scRNA-seq of the adult $Omp^{Cre}$/R26AP-2ε allowed us to understand whether sustentacular cells with ectopic *Tfap2e* expression was contributing to the disorganization of the VNE. By performing differential gene expression analysis on the *Tfap2e* positive and negative sustentacular cells from the adult $Omp^{Cre}$/R26AP-2ε mice, we observed significantly dysregulated genes (550 upregulated; 571 downregulated, adjusted p-value <0.05) with enrichment of genes related to tight junctions, cell-cell adhesion, and cytoskeletal organization (*Figure 5—figure supplement 1*), which may contribute to the disorganized neuroepithelium.

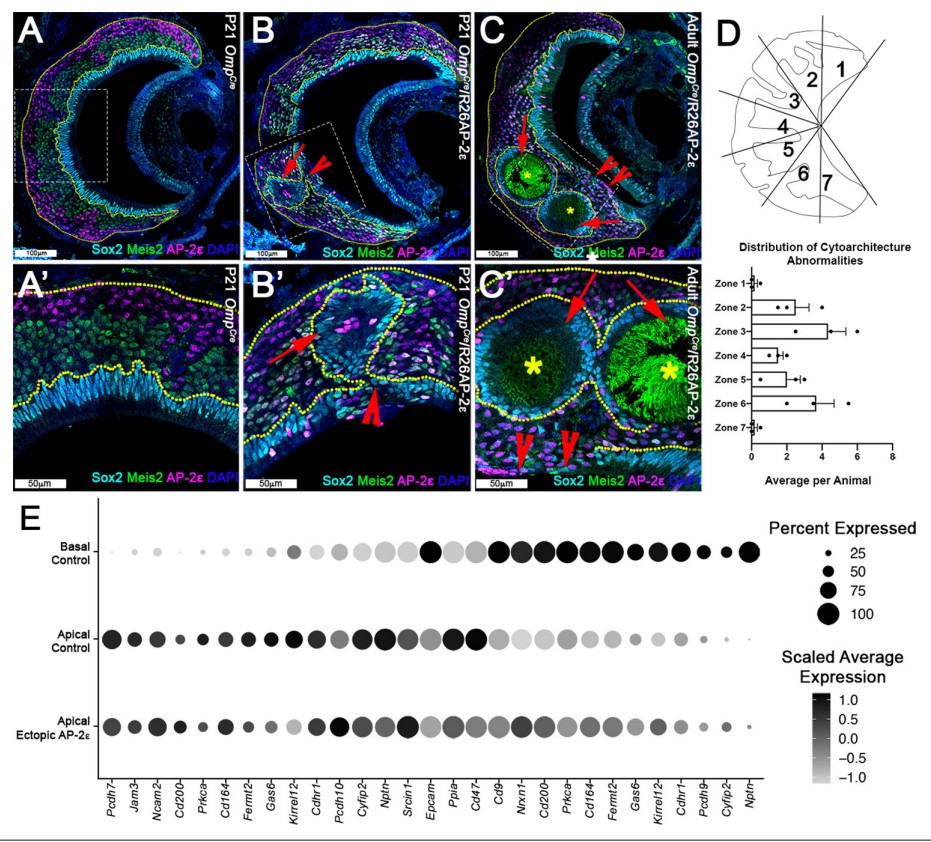

**Figure 5.** Progressive changes in vomeronasal epithelium (VNE) lamination may reflect the changing expression profiles of cell adhesion molecules in apical vomeronasal sensory neurons (VSNs). Immunofluorescence against Sox2 (cyan), AP-2ε (magenta), and Meis2 (green) with 4',6'-diamidino-2-phenylindole (DAPI) (blue) counterstain. Neuroepithelium traced in yellow dotted line. (**A**) P21 $Omp^{Cre}$ controls show highly organized stratified neuroepithelium with contiguous layers of AP-2ε+/basal (magenta), Meis2+/apical (green), and Sox2+/ sustentacular cell (cyan) layers. (**B–C'**) $Omp^{Cre}$/R26AP-2ε vomeronasal organ (VNO) at (**B**) P21 show that Sox2+/ sustentacular cells have intraepithelial cysts with internalized subsidiary lumens (red arrows). (**C**) Adult (3mo) $Omp^{Cre}$/R26AP-2ε mutants show an increase in the severity of intraepithelial cysts (red arrows) and breaks in the sustentacular layer and expansion of neurons to the luminal surface (red notched arrows). Unidentified matter (*) reactive to anti-mouse Abs was detected within the cysts. (**D**) Quantifications of the zonal distribution through Zone 1 (dorsal) → Zone 7 (ventral) of these cytoarchitecture abnormalities (which include both cell body abnormalities and dendritic disorganization, each point = 1 animal) show that these disruptions occur in the intermediate and central regions of the VNO, but not in the marginal zones (zones 1, 7). The highest rate of occurrence are in zones 3 and 6, which are intermediate regions in the VNO. N=21. (**E**) Dot plot showing the composition and intensity of differentially expressed genes involved in cellular adhesion in the VSNs of controls and ectopic mutants.

The online version of this article includes the following figure supplement(s) for figure 5:

**Figure supplement 1.** Analysis of sustentacular cells.

## pS6 immunostaining reveals that $Omp^{Cre}$/R26AP-2ε mutants have defective response to female urines

Whole male mouse urine activates both V1Rs and V2Rs (***Krieger et al., 1999***), while female odorants mostly activate apical VSNs (***Dudley and Moss, 1999***; ***Kimoto et al., 2005***; ***Norlin et al., 2001***; ***Silvotti et al., 2018***). To determine if $Omp^{Cre}$/R26AP-2ε mice had altered chemodetection, we quantified VSNs' activation after exposure of control and mutant mice to either male- or female-soiled bedding. Brains were collected after 90 min of exposure to the soiled bedding, to allow adequate time for the phosphorylation of the ribosomal protein S6 (pS6) in the VSNs' cell bodies (***Silvotti et al., 2018***). VSNs activation was quantified after immunostaining against pS6 (Ser 240/244) on coronal

sections of the VNO (*Figure 6*). Apical and basal VSNs were identified with immunostaining against Meis2 and categorized as either pS6+/Meis2+ apical or pS6+/Meis2- basal VSNs (*Figure 6A–D*). When exposed to male bedding we observed that *Omp^Cre^*/R26AP-2ε mice had a lower average number of activated apical VSNs compared to controls, however this difference was non-statistically significant (*Figure 6A, B and E*). Although this difference was non-statistically significant, we did find a significant reduction in the total activation of VSNs in female mutants.

In order to further analyze the activity of apical VSNs, we exposed control and *Omp^Cre^*/R26AP-2ε males and females to female-soiled bedding which mostly activates apical VSNs. This experiment highlighted a dramatic reduction in apical VSNs' activation, as well as total activation of VSNs (pS6+/Meis2+) of mutant mice for both sexes (*Figure 6C, D and F*).

## Ectopic *Tfap2e* enhances intermale aggressive behavior but not preference for opposite sex odorants

To determine whether the aberrant gene expression in apical VSNs could alter VSNs' functions and related social behaviors, we evaluated intermale aggression and odorant preference (*Koolhaas et al., 2013*). By performing resident intruder tests, we showed that the level of intermale aggression of *Omp^Cre^*/R26AP-2ε male mice was on average higher, but not significantly different than controls (p=0.051) (*Figure 6G*). However, among the animals that displayed aggressive behavior we observed significant increase in the number of attacks from WT to *Omp^Cre^*/R26AP-2ε mutants (p=0.0015; WT = 16.00 SE±3.1; ectopic = 43.75 SE±4.8).

The sex urines preference test revealed that both male and female *Omp^Cre^*/R26AP-2ε mutants exhibited preferential interest in opposite sex urines similar to controls (*Figure 6H*). Interestingly, *Omp^Cre^*/R26AP-2ε females displayed much more variability in their individual odor preferences compared to controls, nonetheless the ectopic mutants still retained their preference for opposite sex odorants.

All together these data suggest that ectopic *Tfap2e* expression decreases apical VSNs' functionality (*Figure 6A–F*) likely leading to an increase in aggression behavior but not compromising opposite sex odorants preferences (*Figure 6G and H*). These data are in line with previous findings indicating that loss of apical VSN signal transduction enhances territorial aggression in males without substantial changes in sex odor preferences (*Trouillet et al., 2019*).

## The negative effects of ectopic *Tfap2e* expression in apical neurons functionality is reflected by reduced c-Fos activation in the anterior AOB

To further investigate if *Tfap2e* ectopic expression in apical neurons alters the vomeronasal signal transduction, we analyzed c-Fos activation in the AOB. To do this we analyzed the AOBs of control and *Omp^Cre^*/R26AP-2ε animals exposed to opposite sex-soiled bedding. This analysis revealed that c-Fos activation in the anterior AOB (aAOB) was statistically different only after exposure to female-soiled bedding (*Figure 6I and K*). C-Fos activation in the aAOB of female *Omp^Cre^*/R26AP-2ε mice exposed to male bedding was on average lower than, but not significantly different from controls (*Figure 6J*). These data suggest that ectopic *Tfap2e* expression reduces the functionality of the V1R VSNs projecting to the aAOB but does not alter the functionality of basal VSNs.

## Ectopic expression of *Tfap2e* alters the transcriptional profile of apical neurons

To further elucidate the gene expression changes in apical (*Meis2*+) VSNs after *Tfap2e* expression, we analyzed the UMAPs using scRNA-seq from VSNs in controls and mutant mice. These revealed similar clustering at the stages of neurogenesis and differentiation across genotypes (*Figure 7*). However, control animals showed *Tfap2e* expression was limited to immature-mature basal VSNs (*Figure 7A'–A'''*; *Figure 4A1*). In *Omp^Cre^*/R26AP-2ε mutants, however, *Tfap2e* mRNA was expressed in maturing basal VSNs as well as in maturing and mature apical VSNs (*Figure 7B' and B'''*; *Figure 4D1*). When analyzing the UMAPs, we noticed that in the *Omp^Cre^*/R26AP-2ε mutants, cells along the apical and basal developmental trajectories overlapped to those of controls. However, the maturing and mature apical VSNs of controls and mutants formed non-overlapping clusters (*Figure 7A'''–(A+B)''''*). In fact, the apical VSNs of the ectopic mutants formed a cluster more proximal to the basal VSNs.

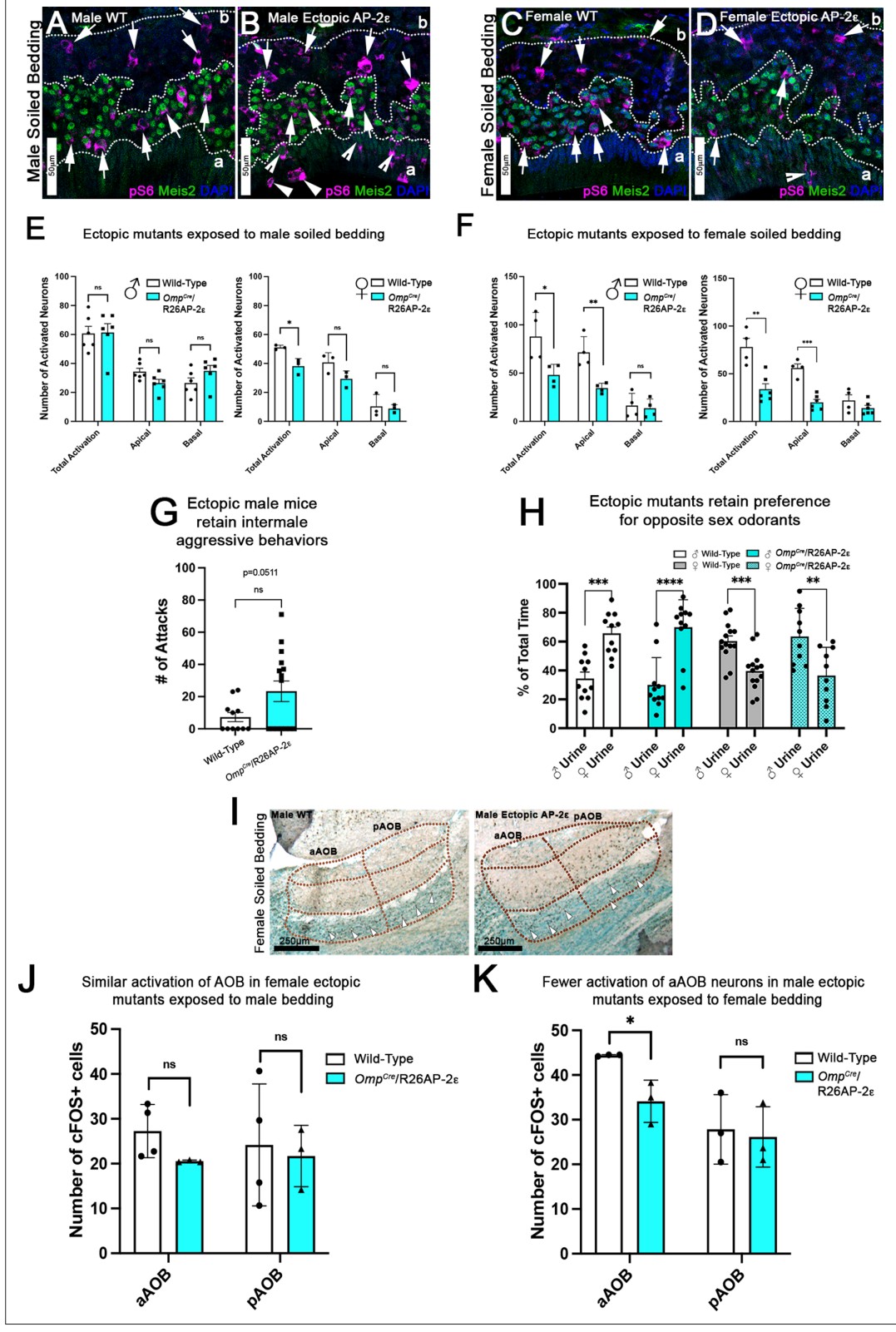

**Figure 6.** Ectopic *Tfap2e* expression alters the detection of sex-specific odorants. (**A–D**) Immunofluorescence against pS6 (magenta, arrows) and Meis2 (green) with 4',6'-diamidino-2-phenylindole (DAPI) counterstain (blue) in the vomeronasal epithelium (VNE) of controls (**A,C**) and *Omp^Cre^*/R26AP-2ε mutants (**B,D**). (**A,B**) VNE of adult male wild-type (WT) and ectopic mutants when exposed to male-soiled bedding show similar activation or pS6

*Figure 6 continued on next page*

*Figure 6 continued*

immunoreactivity (arrows) in Meis2+/apical and Meis2-/basal vomeronasal sensory neurons (VSNs). (**C,D**) VNE of adult female WT and ectopic mutants when exposed to female-soiled bedding show that while WT females displayed a higher proportion of pS6+ apical VSNs, *Omp^Cre^*/R26AP-2ε mutants showed a decreased number of activated Meis2+/apical VSNs. In both exposure conditions ectopic activation of sustentacular cells (notched arrows) and VSNs near the lumen (arrowheads). (**E**) Quantifications of activated VSNs in male and female mice after exposure to male-soiled bedding show a non-significant decrease in activated apical VSNs in mutants of both sexes. Mutant females show a significant decrease in total activation of VSNs (p<0.05) while mutant males show a small but non-significant increase in the total number of activated basal VSNs. Unpaired t-test. Error bars are standard deviation. N=6 for males and n=3 females for both genotypes. (**F**) Quantifications of activated VSNs after exposure to female-soiled bedding in male and female mice show a significant decrease in the number of activated apical VSNs as well as total activation of VSNs between mutants and controls of both sexes (p<0.05 = *, p<0.01 = **, p<0.001 = ***). Unpaired t-test. Error bars are standard deviation. N=4 for males and n=6 for females for both genotypes. (**G**) Quantifications of the number of attacks for all WT and *Omp^Cre^*/R26AP-2ε mutants in a resident intruder test show male ectopic mutant mice display higher levels of intermale aggression but are not significantly different than controls. Unpaired t-test. Error bars are standard deviation. N=11 for WTs and n=17 for *Omp^Cre^*/R26AP-2ε mutants. (**H**) Quantifications of odorant preference tests in male and female mice show that male and female *Omp^Cre^*/R26AP-2ε mutants retain the preference for opposite sex odorants (p<0.05 = *, p<0.01 = **, p<0.001 = ***, p<0.0001 = ****). Unpaired t-test. Error bars are standard deviation. N=11 for WT males, n=14 for WT females, n=11 for *Omp^Cre^*/R26AP-2ε mutant males, and n=10 for *Omp^Cre^*/R26AP-2ε mutant females. (**I**) Accessory olfactory bulb (AOB) of adult male WT and ectopic mutants when exposed to female-soiled bedding show that ectopic mutants have lower levels of cFOS activated neurons in the anterior AOB (aAOB) but similar levels of activated neurons in the posterior AOB (pAOB). (**J**) Quantification of cFOS activated neurons in the AOB of female mice exposed to male-bedding show no significant differences between aAOB and pAOB activation in mutants and WT. Unpaired t-test. Error bars are standard deviation. N=4 for WT and n=3 for *Omp^Cre^*/R26AP-2ε mutants. (**K**) Quantification of cFOS activated neurons in the AOB of male mice exposed to female-soiled bedding show a significant reduction in aAOB activation (p<0.05) in mutants compared to WT. Unpaired t-test. Error bars are standard deviation. N=3 for both WT and *Omp^Cre^*/R26AP-2ε mutants.

The online version of this article includes the following figure supplement(s) for figure 6:

**Figure supplement 1.** Analysis of the accessory olfactory bulbs (AOBs) from adult wild-type (WT) and *Omp^Cre^*/R26AP-2ε mice.

---

To understand the extent to which AP-2ε can reprogram apical VSNs, we further compared the expression of the most enriched genes in apical and basal VSNs of *Omp^Cre^* controls to the apical VSNs of *Omp^Cre^*/R26AP-2ε. Interestingly, this analysis revealed that apical VSNs of *Omp^Cre^*/R26AP-2ε mice had a mixed apical-basal RNA expression profile with a significant downregulation of ~22% of the apical-enriched genes (7/32), and a significant upregulation of ~28% of the basal-enriched genes (20/71) (*Figure 7C*; *Supplementary file 2*). Performing a correlation analysis, we observed that while in controls, sets of either apical- or basal-enriched genes had high correlation, this was no longer true for the *Omp^Cre^*/R26AP-2ε mutants (*Figure 7—figure supplement 1*).

Of the aberrantly expressed genes in the apical VSNs of *Omp^Cre^*/R26AP-2ε mice, we identified a reduction in *Calreticulin* (*Calr*) mRNA levels together with a strong upregulation of *Calreticulin4* (*Calr4*), which persists in adulthood (*Figure 7D–G, Supplementary file 2*). ISH at P11 confirmed that *Calr4* is normally expressed by basal VSNs in controls. However, in *Omp^Cre^*/R26AP-2ε mutants, *Calr4* mRNA was found in both apical and basal VSNs (*Figure 7F and G*). *Calr* is a negative regulator of transport of V2R receptors to the cell membrane (*Dey and Matsunami, 2011*). In line with previous studies (*Dey and Matsunami, 2011*), we found that *Calr* was expressed below ISH detectability. Feature maps also pointed to the ectopic expression of other basal-enriched genes such as *Gnao1*, *Mt3*, and *Fbxo17* in the apical VSNs (*Figure 4*; *Figure 4—figure supplement 3*). In apical cells, these genes are normally either silenced in mature VSNs or absent from the beginning of the differentiation. RNAscope analysis for *Mt3* confirmed ectopic expression in apical cells (*Figure 4—figure supplement 3*).

## *Omp^Cre^*/R26AP-2ε ectopic mutants have normal axonal projections to the AOB

Axonal projection along the anterior-posterior axis of the AOB is largely determined by axon guidance molecules such as Nrp2, Robo2 while the coalescence of VSN axons into glomeruli is largely dictated

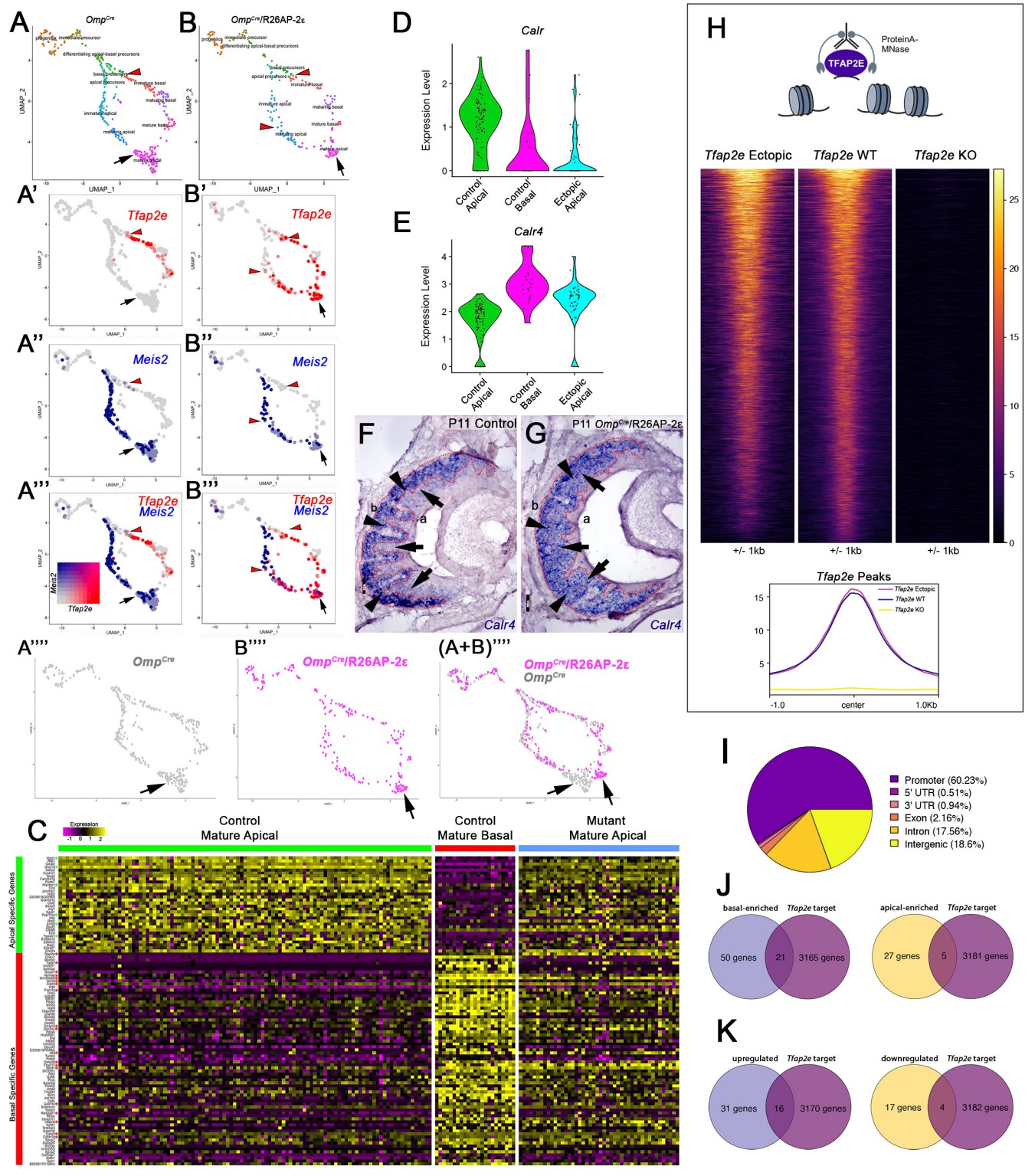

**Figure 7.** Single-cell sequencing of P10 *Omp^Cre* control and *Omp^Cre*/R26AP-2ε mutant vomeronasal sensory neurons (VSNs) indicate a shift in apical cells toward basal cells in the mutant. (**A–A'''**) Uniform Manifold Approximation and Projection (UMAP) clustering of VSNs from progenitor cells to differentiated mature apical and basal cells of control. (**B–B'''**) Mutant mice split by genotype. (**A'–A''', B'–B'''**) Blended feature plots of *Tfap2e* expression (red) and *Meis2* (blue). Red arrowheads indicate the onset of *Tfap2e* expression. Black arrow indicates mature apical VSNs. *Omp^Cre* controls (**A'–A'''**) show a divergent pattern of expression where the onset of *Tfap2e* (red, red arrowhead) is only on the basal branch. *Meis2* expression (blue) occurs only on the apical branch where the cells lack *Tfap2e* expression. *Omp^Cre*/R26AP-2ε mutants (**B'–B'''**) start to express *Tfap2e* on the basal branch

*Figure 7 continued on next page*

*Figure 7 continued*

in immature basal VSNs (red) however, onset of *Tfap2e* mRNA expression also occurs on the apical branch (blue). (**B‴**) In ectopic mutants, *Tfap2e* mRNA is co-expressed with *Meis2* in apical VSNs (purple cells). (**A‴′**) Feature plot of apical cells in *Omp^Cre^* controls (black arrow). (**B‴′**) Feature plot of apical cells in *Omp^Cre^/R26AP-2ε* mutants (black arrow). (A+B‴′) Overlay shows that apical cells with ectopic *Tfap2e* expression (magenta, black arrow) clustered separately from mature apical cells of *Omp^Cre^* controls (gray, black arrow). (**C**) Heatmap showing up/downregulated genes in mature apical and basal VSNs of *Omp^Cre^* controls and in apical VSNs of *Omp^Cre^/R26AP-2ε* mutants. Apical VSNs of ectopic mutants express genes enriched in both apical and basal VSNs. Gene names and values are available in **Supplementary file 1**, **Supplementary file 2**. (**D**) Violin plot shows *Calreticulin* (*Calr*) mRNA expression levels in apical VSNs from the *Omp^Cre^/R26AP-2ε* mutants are reduced to levels similar to basal VSNs from *Omp^Cre^* controls. (**E**) Violin plot shows *Calreticulin-4* (*Calr4*) mRNA expression levels. (**E**) In control cells, *Tfap2e* mRNA expression (red arrowheads) and *Meis2* mRNA expression (blue) are not co-expressed in the same cells. *Tfap2e* expression is upregulated in immature basal VSNs and not apical VSNs. (**F,G**) In situ hybridization (ISH) against *Calr4* against P11 *Omp^Cre^* controls (**F**) and *Omp^Cre^/R26AP-2ε* mutants (**G**) show that while *Calr4* mRNA is normally enriched in basal VSNs (arrowheads), ectopic mutants show expansion of *Calr4* positivity in the apical regions when compared to controls (arrows). (**H–K**) Analysis of CUT&RUN against AP-2ε. (**H**) Tornado plot of AP-2ε occupancy in *Tfap2e* ectopic, wild-type (WT) and knockouts (KOs) in the dissociated tissue of the vomeronasal organ (VNO). AP-2ε signal in ectopic mutants and WTs is similar across all AP-2ε peaks, while *Tfap2e* KOs show no signal. The genomic regions are defined as the summit±1 kb. (**I**) Pie chart depicting the genomic distribution of putative AP-2ε binding sites show that most of AP-2ε peaks are found in promoter regions of putative target genes and to a lesser extent in intergenic and intronic regions of the genome. (**J**) Venn diagram of the determined AP-2ε targets and the genes enriched in the basal and apical VSNs. (**K**) Venn diagram of the determined AP-2ε targets and all the upregulated and downregulated genes in the ectopic mutant mouse. Significance defined as adjusted p-value ≤0.05.

The online version of this article includes the following figure supplement(s) for figure 7:

**Figure supplement 1.** Gene expression correlation changes after ectopic *Tfap2e* expression.

**Figure supplement 2.** CUT&RUN analysis of *Tfap2e* ectopic, wild-type (WT) and knockout (KO) mice.

**Figure supplement 3.** Differential gene expression analysis from single-cell RNA sequencing (scRNA-seq) in adult animals of apical and basal populations scRNA-seq in adults wild-type (WT) controls and *Omp^Cre^/R26AP-2ε* vomeronasal organs (VNOs).

by Kirrel adhesion molecules (***Cloutier et al., 2002***; ***Prince et al., 2013***; ***Prince et al., 2009***; ***Vaddadi et al., 2019***). The mRNA expression levels for the guidance receptors, *Robo2* and *Nrp2*, and the adhesion molecules, *Kirrel2* and *Kirrel3*, did not significantly change after ectopic *Tfap2e* expression. In fact, by immunostaining against Robo2 and Nrp2, we confirmed immunoreactivity of Nrp2 in the aAOB and Robo2 in the posterior AOB (pAOB) similar to controls and observed no significant differences in the average size of aAOB or pAOB between genotypes (***Figure 6—figure supplement 1***). Moreover, quantifications based on Kirrel2 and Kirrel3 immunostaining did not reveal major changes in glomerular size or number in the AOB (***Figure 6—figure supplement 1***; ***Bahreini Jangjoo et al., 2021***).

## Identification of direct AP-2ε targets via CUT&RUN

Our findings suggest a key role for AP-2ε in controlling the expression of specific basal specific/enriched genes. Transcriptomic studies in *Tfap2e^Null^* mice showed loss of expression of basal VSN-specific genes suggesting that AP-2ε controls parts of the basal and apical VSN genetic programs (***Lin et al., 2018***). However, it remains unknown whether AP-2ε regulates VSN genetic programs directly or indirectly. So, we performed genome-wide mapping of TF occupancy with cleavage under targets and release using nuclease (CUT&RUN) to determine the direct genetic targets of AP-2ε in the VNO to pair with our scRNA-seq (***Figure 7H–K***). Our analyses identified over 5025 replicable peaks in VNO tissue indicating AP-2ε binding sites. Notably, performing CUT&RUN and sequencing from *Tfap2e* KOs revealed 203 peaks, of which 154 overlapped with called WT peaks. After subtracting out peaks called in the KO, we were left with 4871 peaks (***Figure 7H***). AP-2ε peaks of the WT were assigned to 3186 genes. None of the peaks of the *Tfap2e* KOs that were subtracted out were associated with apically or basally enriched genes. CUT&RUN peaks of ectopic mutants were largely similar to that of WTs (***Figure 7H***).

Of these putative binding sites, we found that 60.23% of the peaks occurred in promoter regions (defined as any region 1000 bp upstream or 200 bp downstream a transcription start site), 18.6% in distal intergenic regions, and ~17.56% in intronic regions (***Figure 7I***). These results suggest that most AP-2ε activity directly regulates transcription, with perhaps a secondary role in enhancer regions.

Gene Ontology (GO) analysis of genes associated with AP-2ε peaks showed an enrichment of factors involved in protein degradation, transcription coregulator activity, and histone and chromatin modification (***Figure 7—figure supplement 2***). Motif enrichment analysis of AP-2ε peaks revealed

Tfap2 as the top enriched motif (p=1e-180), as expected. Other TF motifs enriched in the same regions as AP-2ε peaks include *Sp, Klf, Ebf, Rfx, Nrf, Dlx, and Lhx* TF families (*Figure 7—figure supplement 2*). As many TFs work with other cofactors to regulate gene expression (*Huang et al., 2015*; *Monahan et al., 2017*), these motifs may represent potential cofactors that work in concert with AP-2ε to mediate either an activating or repressive role in the VNO.

When we compared AP-2ε direct targets with our identified apical- and basal-enriched genes from mature VSN populations, we discovered that 18% of our identified apical-enriched genes (5/27 genes) (*Supplementary file 1*) were AP-2ε direct targets and approximately 47% of our identified basal-enriched genes (21/50 genes) are AP-2ε direct targets (*Figure 7J*). Of the most canonical apical and basal markers and signal transduction machinery, only Gαi2/*Gnai2* had putative direct AP-2ε occupancy and assumed regulation of transcription (*Figure 7—figure supplement 2*). Out of our newly discovered list of basal-enriched genes (*Supplementary file 1*), we identified that *Krt18* has a putative AP-2ε binding site within its promoter region. As expected from a terminal selector gene, these data suggest that AP-2ε directly binds and regulates batteries of apical- and basal-enriched genes (*Figure 7—figure supplement 2*).

These data suggest that *Tfap2e* plays a dual role in maintaining the basal VSNs' genetic program while restricting the expression of genes normally enriched in apical VSNs (*Figure 7J and K*; *Figure 7—figure supplement 2*). In line with this, CUT&RUN from ectopic expressors gave tracks that largely overlapped with those of the WT controls (*Figure 7H*; *Figure 7—figure supplement 2*). However, when plotting the signals at the promoter of apical- and basal-enriched genes for WT, *Tfap2e* KO, and *Tfap2e* ectopic expressors, we observed more signal at the apical-enriched promoters in the ectopic mice dataset (*Figure 7—figure supplement 2*). This data suggests that ectopic expression of *Tfap2e* in apical neurons facilitates its access to the promoter of apical-enriched genes as these are normally active in the apical neurons.

## Discussion

Understanding how differentiated neurons retain cellular plasticity remains critical to identify how genetic insults can compromise neuronal identity, circuit assembly, and function (*Hobert and Kratsios, 2019*; *Molyneaux et al., 2007*; *Patel and Hobert, 2017*; *Pereira et al., 2019*; *Rahe and Hobert, 2019*). Spatial and temporal expression of terminal selector genes regulates that the establishment and maintenance of neuronal identity remains foundational to elucidate the assembly of functional neuronal circuits (*Arlotta et al., 2005*; *Cau et al., 2002*; *Cau et al., 1997*; *Molyneaux et al., 2007*). In fact, loss of terminal selector genes can lead to loss of neuronal identity and increase cellular/phenotypic plasticity, while expression of specific TFs can induce specific cellular features only at particular developmental windows (*Hobert, 2008*; *Rahe and Hobert, 2019*).

Rodents and some marsupials have a binary VNE where the two main types of VSNs, apical and basal VSNs, are generated throughout life from a common pool of Ascl1 progenitors (*Berghard and Buck, 1996*; *Jia and Halpern, 1996*; *Katreddi and Forni, 2021*; *Mohrhardt et al., 2018*; *Silva and Antunes, 2017*; *Taroc et al., 2020*; *Weiler et al., 1999*). The generation of these two distinct populations is central for critical socio-sexual behavior in rodents (*Oboti et al., 2014*; *Pérez-Gómez et al., 2014*). In a recent study we have shown that the apical-basal differentiation dichotomy of VSNs is dictated by Notch signaling (*Katreddi et al., 2022*). *Tfap2e* is expressed in maturing cells fated to become V2R neurons. In this study, we combined scRNA-seq, histology, behavior, and CUT&RUN methodologies to test if *Tfap2e* is a basal VSN-specific terminal selector gene capable of partially reprogramming the apical VSN identity.

Using scRNA-seq, we discovered key transcriptomic differences between mature basal and apical VSNs that were previously unreported (*Figure 1Q*, *Supplementary file 1*). We also confirmed that *Tfap2e* mRNA is restricted to maturing Gαo/basal VSNs (*Figure 1K*) and that AP-2ε itself does not initiate the basal VSN differentiation program, but rather maintains the integrity of the basal neuronal identity. In fact, by re-expressing *Tfap2e* in *Tfap2e* KOs, we demonstrated that AP-2ε is indispensable for basal cellular homeostasis (*Figure 2*) and therefore for the establishment of normal territorial and sex-preference behaviors of rodents (*Figure 3*). We also elucidated that AP-2ε acts in controlling VSN gene expression through activating and repressive activity when analyzing mature/maturing Meis2+ apical VSNs in *Omp^Cre*/R26AP-2ε mutant mice (*Figure 7*).

During differentiation, chromatin barriers dynamically restrict the cellular plasticity, preventing ectopic terminal selector genes from genetically reassigning neurons (*Rahe and Hobert, 2019*). We have previously shown that postmitotic VSNs of *Tfap2e* KO mice can partially deviate from the basal differentiation program and turn on sets of apical-specific genes. However, AP-2ε LOF did not prevent basal neurons from acquiring basal features, such as Gαo or V2Rs (*Lin et al., 2018*), suggesting that AP-2ε activity is crucial in restricting basal VSN phenotypic plasticity rather than establishing the basal cell fate (*Lin et al., 2018*). Here, we showed that *Tfap2e^Null^* mice have reduced odorant sex preference and intermale aggressive behavior, which are classic phenotypes related to basal VSN LOF (*Stowers et al., 2002*). However, we found that reintroducing *Tfap2e* in maturing basal *Tfap2e* KO neurons was sufficient to rescue cellular homeostasis (*Figure 2*), physiological functions, and related behavior (*Figure 3*).

Terminal selectors define neuronal identity by suppressing alternative programs and can also act as pioneer factors (*Lupien et al., 2008*; *Magnani et al., 2011*; *Mangale et al., 2008*). Based on our rescue data, we propose that AP-2ε can partially reprogram/alter the transcriptome of differentiated cells, as expected from a pioneer factor and other members of the Tfap2 family (*Rothstein and Simoes-Costa, 2020*).

When we used *Omp^Cre^* drivers to induce ectopic *Tfap2e* in differentiated olfactory and both apical and basal vomeronasal neurons, we observed progressive gene expression, and morphological changes in the VNO, but no gross morphological changes in the main OE (*Figures 4 and 5*; *Figure 4—figure supplement 2*). We suspect that the lack of phenotype in the OE may arise from the absence of necessary Tfap2 cofactors, which are expressed in the VNO (*Figure 4—figure supplement 1*; *Eckert et al., 2005*).

The co-expression of *Tfap2e* in *Meis2*+ apical VSNs in *Omp^Cre^*/R26AP-2ε mice revealed that Meis2, and most apical-specific genes, were expressed at P10 (*Figure 7C*). However, single-cell transcriptome analyses in adult *Tfap2e* ectopic mice indicated that several apical genes, including *Meis2* and *Calr*, were expressed at significantly lower levels than controls (*Figure 7—figure supplement 3*). These data suggest that AP-2ε can negatively modulate genes enriched in the apical program.

Ectopic expression of individual terminal selector genes can selectively control specific molecular features linked to neuronal function and identity, but not pan-neuronal features like guidance cue receptors (*Patel and Hobert, 2017*; *Stefanakis et al., 2015*). Our scRNA-seq revealed that *Tfap2e* ectopic expression does not alter the expression of VSN-specific guidance cue receptors *Nrp2* and *Robo2* (*Cho et al., 2011*; *Cloutier et al., 2002*; *Prince et al., 2009*; *Walz et al., 2002*). Notably *Nrp2* and *Robo2* start to be expressed soon after the apical/basal VSNs' developmental trajectories are established, suggesting that these genes are expressed before and independently of *Tfap2e* expression (*Figures 1 and 7*; *Figure 6—figure supplement 1*). In line with this, we observed that *Tfap2e* ectopic expression did not significantly change Kirrel2 and Kirrel3 expression patterns (*Figure 6—figure supplement 1*). As a result, we found no significant changes in glomeruli size or number in the AOB (*Figure 6—figure supplement 1*).

In *Omp^Cre^*/R26AP-2ε mutants, we observed that the cellular organization and lamination of the VNE became severely disrupted between P10 and adult ages (*Figure 5*). In fact, in the mutants, we found basal neurons located at the level of the VNE lumen and ectopic sustentacular cells forming intraepithelial cyst-like structures in both apical and basal territories (*Figure 5B and C*). Notably, the disorganization of the VNE that resembles intraepithelial cysts as previously described in aging mice (*Wilson and Raisman, 1980*) appeared to be more pronounced/frequent in regions proximal to the neurogenic marginal zones (*Figure 5D and E*), where *Omp* mRNA is expressed following *Gap43* expression. Therefore, we posit that the regionalization of the VNE phenotypes might represent cells that underwent *Tfap2e* ectopic expression at early maturation stages. When we compared mRNA of control and ectopic mutants, which have disorganized VNE, we observed changes in expression levels of many surface and cell adhesion-related molecules as well as upregulation of stress-related genes (*Figure 5E*; *Figure 5—figure supplement 1*). In addition to this, scRNA-seq analysis of *Tfap2e* positive and negative sustentacular cells in adult *Omp^Cre^*/R26AP-2ε mutants revealed massive changes in gene expression in these support cells. Future studies should focus on understanding which of the dysregulated genes in VSNs and sustentacular cells contribute to the cytoarchitectural organization of VSNs and sustentacular cells.

Sex odorants activate different sets of VRs and therefore different populations of VSNs (**Dudley and Moss, 1999**; **Keller et al., 2006**; **Silvotti et al., 2018**). Interestingly, scRNA-seq analysis and validation via RNA scope and ISH revealed that ectopic *Tfap2e* in mature apical VSNs leads to the upregulation of basal-enriched genes such as *Gnao1* (**Figure 4**) and the ER chaperone protein *Calreticulin-4* (*Calr4*) (**Figure 7**). In the ectopic mutants, we observed a reduction in mRNA of *Calr* (**Figure 7D**). Loss of *Calr* expression has been previously shown to increase V2R cell surface expression (**Dey and Matsunami, 2011**). Changes in *Calr4* and *Calr* expression levels could be partially responsible for the overall increase in family-C V2Rs cell surface expression/immunoreactivity (**Dey and Matsunami, 2011**) observed in ectopic mutants (**Figure 4C, F and H**).

After *Omp^{Cre}*-driven ectopic *Tfap2e* expression, we observed a significant upregulation of mRNAs of V2R receptors coded by genes belonging to the C-family (**Silvotti et al., 2011**; **Figure 4—figure supplement 2**). Notably, our data suggest that some apical neurons of controls can express low mRNA levels of family-C *Vmn2r* mRNAs (**Figure 4**; **Figure 4—figure supplement 2**). These data suggest that some family-C genes (e.g. *Vmn2r7*), which are known to not follow rigorous mechanisms of monogenic expression, might also have a much looser cell type-specific expression control than previously postulated. Differential transcriptome analysis between controls and *Omp^{Cre}*/R26AP-2ε mutants did not reveal any additional differences in V1R genes and other V2R genes across genotypes. However, due to the small representation of each single VR gene across neurons and the relatively small number of cells sequenced, we could not arrive to definitive conclusions in this regard.

Using the anti-V2R2 antibody, which recognized a large spectrum of Vmn2r of the family-C, including Vmn2r7 (**Silvotti et al., 2011**), we could detect cells immunoreactive for both V2R2 and Meis2 in ectopic mutants (**Figure 4L**). This suggests that in control animals apical VSNs either express V2R genes below immune detectability (**Figure 4K**) or that post-transcriptional mechanisms may play roles in silencing translation.

Using pS6 immunoreactivity to detect activated VSNs, and cFos to detect signal transduction in the AOB revealed that ectopic mutants have an overall decreased apical cells' ability to detect and transduce signals to the aAOB (**Figure 6**). Notably these defects resulted more obvious, in both males and females, after female bedding exposure (**Figure 6**).

Although the transcriptome profile of apical neurons in *Omp^{Cre}*/R26AP-2ε mutants is changed (**Figure 7**; **Supplementary file 2**), we did not observe significant changes in V1R gene expression. As many genes seem to be mis-regulated, it is tempting to speculate that the reduced response of apical neurons to stimuli is a result of changes in post-transcriptional or post-translational modification, trafficking of the receptors, or even competition between *Gnai2* and *Gnao1*. Conditional ablation of normal apical VSN signal transduction does not undermine intermale aggression, rather it enhances territorial aggression in mutant males (**Trouillet et al., 2019**). In line with this, we found that *Omp^{Cre}*/R26AP-2ε mutants displaying aggressive behavior had higher levels of aggression toward male intruders when compared to WT controls (**Figure 6G**). These data indicate that ectopic *Tfap2e* expression in V1R neurons is sufficient to partially subvert their function and therefore alter intrinsic social behaviors.

In rodents, olfactory sex discrimination persists after VNO excision; however, preference for opposite sex odorants is mediated by the AOS (**Keller et al., 2006**; **Pankevich et al., 2004**). Despite the partial desensitization of the apical VSNs in *Omp^{Cre}*/R26AP-2ε mutants, transcriptome and morphological changes in the VNO did not compromise normal sex odorant preference in male or female mutants. In fact, conditional ablation of Gαi2, which is required for normal signaling of apical VSNs, did not alter normal male sexual behavior, including male preference for estrous female urine (**Trouillet et al., 2019**). Our data support a dispensable role for apical VSNs neurons in the tested sexual behaviors (**Figure 6C, D and H**).

To elucidate the mechanism of action of AP-2ε, we performed CUT&RUN (**Skene et al., 2018**; **Skene and Henikoff, 2017**). This analysis (**Figure 7**) identified putative direct targets of AP-2ε and showed 3000+ putative binding sites in the vomeronasal tissue. The specificity of binding was confirmed by CUT&RUN experiments on *Tfap2e* KOs (**Figure 7H**). In the identified AP-2ε target genes, we found that most of the binding sites were primarily in promoter regions, not in intergenic and intronic regions. These data suggest that AP-2ε's main mechanism of action directly regulates gene transcription with perhaps a secondary role at enhancer regions. We found that AP-2ε bound to the up- and downregulated genes in both apical VSNs and sustentacular cells in the *Omp^{Cre}*/R26AP-2ε ectopic

mouse line, which suggests that AP-2ε can act as both a transcriptional activator and repressor. Motif analysis of these up- and downregulated regions indicates that AP-2ε may function in concert with specific transcriptional cofactors to fulfill a dual role in maintaining the basal VSN genetic program and restricting cellular plasticity (*Figure 7—figure supplement 2*). Notably our CUT&RUN data showed an enrichment of factors involved in transcription coregulator activity, histone modification, and chromatin modification (*Figure 7—figure supplement 2*). These data indicate that AP-2ε may play a role in modifying the chromatin landscape indirectly or in tandem with these transcriptional cofactors to regulate the basal genetic program. Even though transcriptome and histological analysis of the VNE showed significant changes in canonical apical and basal specific genes, we only had *Gnai2* and *Krt18* AP-2ε peak assignments, suggesting that AP-2ε acts indirectly to regulate these genes. However, since peaks were assigned to the nearest gene, we cannot exclude long-distance gene regulation through enhancer regions as a contributor. CUT&RUN peaks of ectopic mutants were largely similar to that of WTs, while only background peaks were found in *Tfap2e* KOs. Notably, our data suggests that ectopic expression of *Tfap2e* in apical neurons allows for access of AP-2ε to the promoter of apical-enriched genes as they are in the euchromatic state (*Figure 7—figure supplement 2*).

In conclusion, the results of our study indicate that *Tfap2e* has some features of a terminal selector gene. In fact, *Tfap2e* plays roles in controlling the expression or expression levels of several basal-enriched VSN genes and repressing apical ones, which is necessary for normal basal VSN functions that mediate territorial and sex preference behaviors in mice. We recently found that the establishment of the apical/basal identity is a slow and multistep process, and that the apical/basal identity is primarily established by Notch signaling as soon as the cells become postmitotic (*Katreddi et al., 2022*). In line with this, we observed that after ectopic *Tfap2e* expression in maturing apical neurons the core of their default apical identity is maintained, however some of the genes normally expressed in alternative basal identity are turned on in apical neurons. Our data suggest that after *Tfap2e* expression, the apical VSNs acquire an ambiguous transcriptome identity as ectopic expression is sufficient to bypass some layers of cellular plasticity restrictions over time. These changes translate into a reduced functionality of the apical neurons, rather than into a transdifferentiation to basal VSNs. The genetic changes induced by ectopic *Tfap2e* further manifest in a progressive disorganization of the vomeronasal neuroepithelium like that reported in aging animals (*Wilson and Raisman, 1980*).

Our study suggests that as previously hypothesized by others (*Hobert and Kratsios, 2019*; *Rahe and Hobert, 2019*), aberrant expression of terminal selector genes in postnatal neurons can alter the transcriptomic identity of neurons, organization neuroepithelia, and potentially lead to neuropathologies.

# Materials and methods

## Key resources table

| Reagent type (species) or resource | Designation | Source or reference | Identifiers | Additional information |
|---|---|---|---|---|
| Antibody | Anti-gao (Rabbit polyclonal) | Invitrogen | PA5-59337 | (1:1000) |
| Antibody | Anti-gao (Mouse monoclonal) | Synaptic Systems | 271 111 | (1:250) |
| Antibody | Anti-gai2 (Mouse monoclonal) | Millipore | MAB3077 | (1:250) |
| Antibody | Anti-keratin 18 (Rabbit polyclonal) | Abcam | ab52948 | (1:500) |
| Antibody | Anti-kirrel2 (Goat polyclonal) | R&D Systems | AF2930 | (1:500) |
| Antibody | Anti-kirrel3 (Mouse monoclonal) | NeuroMab | 75–333 | (1:100) |
| Antibody | Anti-meis2 (Mouse monoclonal) | Santa Cruz | sc-81986 | (1;500) |
| Antibody | Anti-meis2 (Rabbit polyclonal) | Abcam | ab73164 | (1:1000) |
| Antibody | Anti-Nrp2 (Goat polyclonal) | R&D Systems | AF567 | (1:4000) |
| Antibody | Anti-olfactory marker protein (Goat polyclonal) | WAKO | 54410001 | (1:4000) |

*Continued on next page*

*Continued*

| Reagent type (species) or resource | Designation | Source or reference | Identifiers | Additional information |
|---|---|---|---|---|
| Antibody | Anti-phospho-S6 Ribosomal Protein (Ser240/244) (Rabbit monoclonal) | Cell Signaling Technology | D68F8 | (1:500) |
| Antibody | Anti-robo2 (Mouse monoclonal) | Santa Cruz | sc-376177 | (1:250) |
| Antibody | Anti-sox2 (Goat polyclonal) | R&D Systems | AF2018 | (1:800) |
| Antibody | Anti-AP-2ε (goat polyclonal) | R&D Systems | AF5060 | (1:500) |
| Antibody | Anti-AP-2ε (Rabbit polyclonal) | ProteinTech Group | 25829–1-AP | (1:500) |
| Antibody | Anti-c-Fos (Rabbit monoclonal) | Abcam | ab53212 | (1:500) |
| Antibody | V2R2 (Rabbit polyclonal) | Roberto Tirindelli (University of Parma) | | (1:5000) |
| Antibody | Anti-goat (Alexa488) (Donkey polyclonal) | Molecular Probes | a11055 | (1:1000) |
| Antibody | Anti-goat (Alexa594) (Donkey polyclonal) | Molecular Probes | a11058 | (1:1000) |
| Antibody | Anti-goat (biotinylated) (Horse polyclonal) | Vector | ba9500 | (1:1000) |
| Antibody | Anti-mouse (Alexa488) (Donkey polyclonal) | Molecular Probes | a21202 | (1:1000) |
| Antibody | Anti-mouse (Alexa594) (Donkey polyclonal) | Molecular Probes | a10037 | (1:1000) |
| Antibody | Anti-mouse (biotinylated) (Horse polyclonal) | Vector | ba2000 | (1:1000) |
| Antibody | Anti-rabbit (Alexa488) (Donkey polyclonal) | Molecular Probes | a21206 | (1:1000) |
| Antibody | Anti-rabbit (Alexa594) (Donkey polyclonal) | Jackson Labs | 711-585-152 | (1:1000) |
| Antibody | Anti-rabbit (biotinylated) (Horse polyclonal) | Vector | ba1100 | (1:1000) |
| Sequence-based reagent | RNAscope Probe-Mm-Abca7-C4 | Acdbio | 489021 | |
| Sequence-based reagent | RNAscope Probe-Mm-Gnai2 | Acdbio | 868051 | |
| Sequence-based reagent | RNAscope Probe-Mm-Gnao1-E4-E6-C2 | Acdbio | 444991 | |
| Sequence-based reagent | RNAscope Probe-Mm-Mt3-C3 | Acdbio | 504061 | |
| Sequence-based reagent | RNAscope Probe-Mm-Meis2-C3 | Acdbio | 436371 | |
| Chemical compound, drug | Neuron Isolation Enzyme (with papain) | Thermo Fisher Scientific | 88285 | |
| Chemical compound, drug | Neurobasal Medium | Thermo Fisher Scientific | 21103049 | |
| Chemical compound, drug | Dimethyl sulfoxide (DMSO) | Sigma-Aldrich | 472301 | |
| Chemical compound, drug | Diaminobenzidine (DAB) | Sigma | D9015-100MG | (250 µg/ml) |
| Chemical compound, drug | DAPI | CALBIOCHEM | 268298 | (1:3000) |
| Commercial assay or kit | DIG Labelling Kit | Roche | 11 175 025 910 | |

*Continued on next page*

*Continued*

| Reagent type (species) or resource | Designation | Source or reference | Identifiers | Additional information |
|---|---|---|---|---|
| Commercial assay or kit | DIG Detection Kit | Roche | 11 175 041 910 | |
| Chemical compound, drug | 37% Formaldeyde | Sigma | F1635-500ML | |
| Chemical compound, drug | Tissue-Tek O.C.T. Compound | VWR | 25608-930 | |
| Commercial assay or kit | ABC HRP Kit | Vector | PK-6100 | |
| Commercial assay or kit | NEBNext Ultra II DNA Library Prep Kit for Illumina | New England Biolabs | E7645 | |
| Commercial assay or kit | RNAscope Intro Pack for Multiplex Fluorescent Reagent Kit v2- Mm | Acdbio | 323136 | |
| Commercial assay or kit | RNAscope 4-Plex Ancillary Kit for Multiplex Fluorescent Kit v2 | Acdbio | 323120 | |
| Biological sample (*Mus musculus*) | *Tfap2e^{Cre}* | Trevor Williams (University of Colorado, Denver) | | |
| Biological sample (*Mus musculus*) | B6;129P2-Omptm4(cre)Mom/MomJ | Jackson Labs | Stock No: 006668 | |
| Biological sample (*Mus musculus*) | B6.Cg-Gt(ROSA)26Sor tm(CAG-mTfap2e) For (R26AP-2ε) | This paper | | Will Be available from Jackson Labs |

| Reagent type (species) or resource | Designation | Source or reference | Identifiers | Additional information |
|---|---|---|---|---|
| Software, algorithm | FIJI ImageJ software (Version 2.1.0/1.53c) | NIH | https://imagej.nih.gov/ij/ | |
| Software, algorithm | GraphPad Prism 9 software | GraphPad | https://www.graphpad.com/ | |
| Software, algorithm | Photoshop CC 2020 | Adobe | https://www.adobe.com/products/photoshop.html | |
| Software, algorithm | RStudio (Version 1.3.1073) | RStudio | https://rstudio.com/ | |
| Software, algorithm | R version 4.0.2 | The R Project | https://www.r-project.org/ | |
| Software, algorithm | Seurat 4.0.5 | Satija Lab | https://satijalab.org/seurat/ | |
| Software, algorithm | ButtonBox v.5.0 | Behavioral Research Solutions, LLC | | Shared by Damien Zuloaga (University at Albany) as Software is discontinued |
| Software, algorithm | Cutadapt | *Martin, 2011* | https://cutadapt.readthedocs.io/en/stable/ | |
| Software, algorithm | Bowtie 2 | *Langmead and Salzberg, 2012* | http://bowtie-bio.sourceforge.net/bowtie2/index.shtml | |
| Software, algorithm | MACS2 | *Zhang et al., 2008* | https://chipster.csc.fi/manual/macs2.html | |
| Software, algorithm | ChIPseeker | *Yu et al., 2015* | https://bioconductor.org/packages/release/bioc/html/ChIPseeker.html | |
| Software, algorithm | clusterProfiler | *Yu et al., 2012* | https://bioconductor.org/packages/release/bioc/html/clusterProfiler.html | |
| Software, algorithm | HOMER | Benner Lab | http://homer.ucsd.edu/homer/ | |

*Continued*

| Reagent type (species) or resource | Designation | Source or reference | Identifiers | Additional information |
|---|---|---|---|---|
| Sequence-based reagent | GGA GGG GGG CTCT GAG AT | This paper | R26-AP-2ε Common | Will Be available from Jackson Labs |
| Sequence-based reagent | GGC TGG TGT GGC CAA TGC | This paper | R26-AP-2ε Mutant | Will Be available from Jackson Labs |
| Sequence-based reagent | GTC GTG AGG CTG CAG GTC | This paper | R26-AP-2ε WT | Will Be available from Jackson Labs |
| Sequence-based reagent | AGT TCG ATC ACT GGA ACG TG | Jackson Labs | *Omp* WT Fwd | |
| Sequence-based reagent | CCC AAA AGG CCT CTA CAG TCT | Jackson Labs | *Omp* WT Rvs | |
| Sequence-based reagent | TAG TGA AAC AGG GGC AAT GG | Jackson Labs | *Omp* Mutant Fwd | |
| Sequence-based reagent | AGA CTG CCT TGG GAA AAG CG | Jackson Labs | *Omp* Mutant Rvs | |
| Sequence-based reagent | GCT GGT GAG TCA ACC TGC CTG CAG | Trevor Williams (University of Colorado, Denver) | AP-2ε WT AK19 | |
| Sequence-based reagent | GGT CAC CTT GTA CTT GGA GTG TGA G | Trevor Williams (University of Colorado, Denver) | AP-2ε WT AK20 | |
| Sequence-based reagent | AGG TGT AGA GAA GGC ACT TAG C | Jackson Labs | Cre Fwd | |
| Sequence-based reagent | CTA ATC GCC ATC TTC CAG CAG G | Jackson Labs | Cre Rvs | |

## Animals

The R26AP-2ε mice were produced by Cyagen (Santa Clara, CA) on a C57B/6 background. The *Tfap2e^Cre* line (*Tfap2e^tm1(cre)Will*) was obtained from Dr Trevor Williams, Department of Craniofacial Biology, University of Colorado. The R26AP-2ε (B6.Cg-Gt(ROSA)26Sor^tm(CAG-mTfap2e)For) mouse line was produced through Cyagen on a *C56BL/6* background. The *Omp^Cre* line (B6;129P2-*Omp^tm4(cre)Mom*/MomJ) was obtained from Dr Paul Feinstein (Hunter College, City University of New York) on a *129P2/OlaHsd* background and backcrossed to a C57BL/6 background for six generations at the time of this study. The characterization and comparison of the rescue of the *Tfap2e* phenotype (*Tfap2e^Cre*/R26AP-2ε), *Tfap2e* KO, and WTs were performed on a C57BL/6 background. *Omp^Cre*/R26AP-2ε mutant mice are viable. Genotyping of mutants was performed by PCR. Primers used are detailed in the Key resources table.

Mice were housed under a 12 hr day/night cycle. Animals were collected/analyzed at P10, P21, and adult (P60-P90) ages. For all morphological analyses both males and females were included unless otherwise specified. All mouse studies were approved by the University at Albany Institutional Animal Care and Use Committee (IACUC). Mouse lines generated in this study will be deposited to Jackson Labs by the time of publication.

## Generation of the Tfap2e conditional knock-in model

The *Tfap2e* conditional knock-in allele was generated by targeting the *Rosa26* gene in C57BL/6 ES cells. The '*CAG-loxP-stop-loxP-mouse Tfap2e CDS-polyA*' cassette was cloned into intron 1 of *Rosa26* in the reverse orientation. In the targeting vector, the positive selection marker (Neo) was flanked by SDA (self-deletion anchor) site, and DTA was used for negative selection. Mouse genomic fragments containing homology arms (Has) were amplified from *BAC* clone by using high fidelity Taq DNA polymerase and were sequentially assembled into a targeting vector together with recombination sites and selection markers.

The *Rosa26* targeting construct was linearized by restriction digestion with *AscI* followed by phenol/chloroform extraction and ethanol precipitation. The linearized vector was transfected into

C57BL/6 ES cells according to Taconic-Cyagen's standard electroporation procedures and G418 resistant clones were selected for 24 hr post-electroporation. These were then screened for homologous recombination by PCR and characterized by Southern blot analysis. Two separate clones, A2 and H2, were successfully transmitted to germline and characterized.

Genotyping for the R26AP-2ε mouse line was performed by PCR using R26-AP-2ε Common (5' GGAGGGGGGCTCTGAGAT 3'), R26-AP-2ε Mutant (5' GGCTGGTGTGGCCAATGC 3'), R26-AP-2ε WT (5' GTCGTGAGGCTGCAGGTC 3') with expected bands at 552 bp (mutant) and 400 bp (WT).

## Both Omp^Cre and WT mice are used as controls depending on availability during performed experiments

### Innate olfactory preference test

Adult mice were isolated for at least 1 week prior to testing. Individual mice were habituated to the experimental environment for at least 30 min, then to the test cage for an additional 2 min. After the habituation period, cotton swabs scented with either male or female whole urine was placed on either side of the test cage. The time spent sniffing each odorant was normalized to total investigation time.

### Resident intruder test

The resident intruder assay was used to evaluate aggression in male mice of mutants and controls. Test subjects were housed with intact females for at least 1 week prior to testing. On the day of testing, all subjects (residents and intruders) were acclimated to the experimental environment for at least 30 min prior to the assay. Females were removed immediately before testing. Castrated C57B mice were swabbed with male whole urine immediately before being introduced into the resident male's home cage. Interactions between isolated residents and intruders were recorded for 10 min and videos were evaluated using ButtonBox v.5.0 (Behavioral Research Solutions, Madison, WI) software for the number and duration of attacks. The same subjects used for innate olfactory preference tests were then used for the resident intruder tests.

## Neuronal activation in response to sex-specific odorants

Adult mice were isolated for at least 1 week prior to exposure to either soiled bedding from male or female mice for ~90 min then perfused with PBS and 3.7% formaldehyde in PBS, then collected to evaluate neuronal activation with immunohistochemistry against pS6.

## Tissue preparation

Tissue collected at ages ≥P10 were perfused with PBS then 3.7% formaldehyde in PBS. Brain tissue was isolated at the time of perfusion and then immersion-fixed for 3–4 hr at 4°C. Noses were immersion fixed in 3.7% formaldehyde in PBS at 4°C overnight and then decalcified in 500 mM EDTA for 3–4 days. All samples were cryoprotected in 30% sucrose in PBS overnight at 4°C, followed by embedding in Tissue-Tek O.C.T. Compound (Sakura Finetek USA, Inc, Torrance, CA) using dry ice, and stored at –80°C. Tissue was cryosectioned using a CM3050S Leica cryostat at 16 µm for VNOs and 20 µm for brain tissue and collected on VWR Superfrost Plus Micro Slides (Radnor, PA) for immunostaining and ISH. All slides were stored at –80°C until ready for staining.

## Immunohistochemistry

For immunohistochemistry and immunofluorescence, antigen retrieval was performed on slides that were submerged in citrate buffer (pH 6.0) above 95°C for at least 15 min before cooling to room temperature, then permeabilized with and blocked in horse serum-based blocking solution before transferring into primary antibodies overnight at 4°C. For immunohistochemistry slides were additionally incubated in an $H_2O_2$ solution (35 mL PBS + 15 mL 100% methanol + 500 µL 30% $H_2O_2$) after antigen retrieval.

For chromogen-based reactions, staining was visualized with the Vectastain ABC Kit (Vector, Burlingame, CA) using diaminobenzidine (*Forni et al., 2011*) sections were counterstained with methyl green and mounted with Sub-X mounting medium. For immunofluorescence species-appropriate secondary antibodies conjugated with either Alexa Fluor 488, Alexa Fluor 594, Alexa Fluor 568, Alexa Fluor 680 were used for immunofluorescence detection (Molecular Probes and Jackson ImmunoResearch Laboratories, Inc, Westgrove, PA). Sections were counterstained with 4',6-diamidino-2-phenylindole

(1:3000; Sigma-Aldrich), and coverslips were mounted with FluoroGel (Electron Microscopy Services, Hatfield, PA).

Confocal microscopy pictures were taken on a Zeiss LSM 710 microscope. Epifluorescence pictures were taken on a Leica DM4000 B LED fluorescence microscope equipped with a Leica DFC310 FX camera. Images were further analyzed using FIJI/ImageJ software. Antibodies and concentrations used in this study are detailed in the Key resources table.

## ISH and RNAscope

Digoxigenin-labeled RNA probes were prepared by in vitro transcription (DIG RNA labeling kit; Roche Diagnostics, Basel, Switzerland). ISH were performed on 16 µm cryosections that were rehydrated in ×1 PBS for 5 min, fixed in 4% PFA in 0.1 M phosphate buffer for 20 min at 4°C, treated with 10 µg/mL proteinase K (Roche) for 12 min at 37°C, and then refixed in 4% PFA at 4°C for 20 min. To inactivate the internal alkaline phosphatase, the tissue was treated with 0.2 M HCl for 30 min. Nonspecific binding of the probe to slides was reduced by dipping slides in 0.1 M triethanolamine (pH 8.0)/0.25% acetic anhydride solution, then washed with ×2 saline-sodium citrate (SSC) buffer before incubating in hybridization solution for 2 hr at room temperature. Slides were then hybridized with 200 µL of probe in hybridization solution at 65°C overnight in a moisture chamber. After hybridization, the slides were washed in ×2 SSC, briefly, then in ×1 SSC/50% formamide for 40 min at 65°C. RNase A treatment (10 µg/mL) was carried out at 37°C for 30 min. The slides were then washed with ×2 SSC then ×0.2 SSC for 15 min each at 65°C. Hybridization was visualized by immunostaining with an alkaline phosphatase conjugated anti-DIG (1:1000), and NBT/BCIP developer solution (Roche Diagnostics). After color reaction, the slides were put into 10 mM Tris-HCl pH 8.0/1 mM EDTA, rinsed in PBS and air-dried before mounting with Sub-X mounting medium. The probe against Gαo (*Gnao1*) was generated as previously described (*Lin et al., 2018*).

Single-molecule fluorescence ISH was performed using the RNAscope Multiplex Fluorescence v2 assay and probes (RNAscope Probe-Mm-Abca7-C4 #489021, RNAscope Probe-Mm-Gnai2 #868051, RNAscope Probe-Mm-Gnao1-E4-E6-C2 #444991, RNAscope Probe-Mm-Mt3-C3 #504061, RNAscope Probe-Mm-Meis2-C3 #436371) from ACDbio. The assay was performed on 16 µm fixed-frozen P10-P11 mouse cryosections, following the manufacturer's protocol.

## Single-cell RNA sequencing

The VNOs of *Omp^Cre* at P10 and *Omp^Cre*/R26AP-2ε at P10 and 3mo were isolated and dissociated into single-cell suspension using neural isolation enzyme/papain (NIE/Papain in Neurobasal Medium with 0.5 mg/mL Collagenase A, 1.5 mM L-cysteine, and 100 U/mL DNAse I) incubated at 37°C. The dissociated cells were then washed with HBSS and reconstituted in cell freezing medium (90% FBS, 10% DMSO). Cells were frozen from room temperature to –80°C at a –1 °C/min freeze rate. Single-cell suspension was sent to SingulOmics for high-throughput single-cell gene expression profiling using the ×10 Genomics Chromium Platform. Data were analyzed along with using Seurat 4.0.5. The scRNA-seq data discussed in this publication have been deposited in NCBI's Gene Expression Omnibus and are accessible through GEO series accession number GSE192746. We also utilized previously published data from *Katreddi et al., 2022*, available through GEO series accession number GSE190330.

## CUT&RUN

Cells frozen in 90% FBS/10% DMSO were thawed at 37°C and resuspended in CUT&RUN wash buffer (20 mM HEPES pH 7.5, 150 mM NaCl, 0.5 mM spermidine, plus Roche Complete Protease inhibitor, EDTA-free). CUT&RUN experiments were performed as previously described (*Meers et al., 2019*) with minor modifications; 0.025% digitonin was used for the Dig-wash buffer formulation. Antibody incubation was performed overnight at 4°C, followed by Protein A-MNase binding for 1 hr at 4°C. Prior to targeted digestion, cell-bead complexes were washed in low-salt rinse buffer (20 mM HEPES pH 7.5, 0.5 mM spermidine, 0.025% digitonin, plus Roche Complete Protease inhibitor, EDTA-free) followed by targeted digestion in ice-cold high-calcium incubation buffer (3.5 mM HEPES pH 7.5, 10 mM CaCl₂, 0.025% digitonin) for 30 min at 0°C. Targeted digestion was halted by replacing the incubation buffer with EGTA-STOP buffer (170 mM NaCl, 20 mM EGTA, 0.025% digitonin, 20 µg/mL glycogen, 25 µg/mL RNase A), followed by chromatin release and DNA extraction. Protein AG-MNase

was kindly provided by Dr Steve Henikoff. A rabbit polyclonal Anti-TFAP2E antibody (Proteintech 25829-1-AP) was used at a concentration of 1:50 for CUT&RUN experiments.

### CUT&RUN library preparation

CUT&RUN libraries were prepared using the NEBNext ultra II DNA library prep kit (New England Biolabs E7645). Quality control of prepared libraries was conducted using an ABI 3730xl DNA analyzer for fragment analysis. Libraries were pooled to equimolar concentrations and sequenced with paired-end 37 bp reads on an Illumina NextSeq 500 instrument.

### Quantification and statistical analyses of microscopy data

All data were collected from mice kept under similar housing conditions in transparent cages on a normal 12 hr light/dark cycle. Tissue collected from either males or females in the same genotype/ treatment group were analyzed together unless otherwise stated. Ages analyzed are indicated in text and figures. The data are presented as mean ± SEM. Prism 9.2.0 was used for statistical analyses, including calculation of mean values, and standard errors. Two-tailed, unpaired t-test were used for all statistical analyses, between two groups, and calculated p-values <0.05 were considered statistically significant. Sample sizes and p-values are indicated as single points in each graph and/or in figure legends.

Measurements of VNE and cell counts were performed on confocal images or bright-field images of coronal serial sections immunostained or ISH for the indicated targets. In animals ≥P15, the most central 6–8 sections on the rostro-caudal axis of the VNO were quantified and averaged, and in animals ≥P0, the most medial 4–6 sections were quantified and averaged. Measurements and quantifications were performed using ImageJ 2.1.0 and Imaris. Statistical differences between two genotypes were quantified with two-tailed unpaired t-test using Prism 9.2.0 (GraphPad Software, San Diego, CA). Microscopy data reported in this paper will be shared by the lead contact upon request.

### Statistical analyses of behavior

Two-tailed unpaired t-test using Prism 9.2.0 (GraphPad Software, San Diego, CA) was used to determine statistical significance between two independent distributions. p-Values <0.05 were considered statistically significant. Sample sizes and p-values are indicated as single points in each graph and/or in figure legends. When performing one-way ANOVA test, Tukey-HSD post hoc comparison was used if statistical significance (p-value <0.05) was determined.

### CUT&RUN data analysis

In processing CUT&RUN data, paired-end sequencing reads were trimmed using Cutadapt t (*Martin, 2011*) using the following arguments: '-a AGATCGGAAGAGCACACGTCTGAACTCCAGTCA -A AGATCGGAAGAGCGTCGTGTAGGGAAAGAGTGT `--minimum-length=25`'. Reads were aligned to the reference mouse mm10 assembly from the UCSC genome browser using Bowtie 2 (*Langmead and Salzberg, 2012*) using the following arguments: '`--local --very-sensitive-local --no-unal --no-mixed --no-discordant -I 10X 1000`'. BAM files were filtered with SAMtools to discard unmapped reads, those which were not the primary alignment, reads failing platform/vendor quality checks, and PCR/optical duplicates (-f 2F 780). Peak calling was performed using MACS2 (*Zhang et al., 2008*). Peak-gene annotation was done by mapping peaks to their closest annotated gene using the ChIPseeker R package (*Yu et al., 2015*). GO term analysis was performed in R using clusterProfiler (*Yu et al., 2012*). Motif enrichment analysis was performed using HOMER (*Heinz et al., 2010*). The data from this CUT&RUN experiment has been deposited into the NCBI's Expression Omnibus and are accessible through GEO series accession number GSE193139.

### Resource availability

#### Lead contact

Further information and requests for resources and reagents should be directed to and will be fulfilled by the lead contact, Paolo E Forni (pforni@albany.edu).

#### Materials availability

Mouse lines generated in this study will be deposited to Jackson Labs by the time of publication.

There are restrictions in availability of the antibody Rabbit anti-V2R2 which was obtained from the lab of Dr Roberto Tornielli (University of Parma, Italy) and is not commercially available.

## Data and code availability
The single-cell RNA-sequencing and CUT& RUN sequencing data discussed in this publication have been deposited at NCBI's Gene Expression Omnibus and are publicly available as of the date of publication. Accession numbers are listed in the Key resources table. This paper reports no original code. Any additional information required to reanalyze the data reported in this paper is available from the lead contact upon request.

# Acknowledgements
This publication was supported by the National Institutes of Health (NIH) by the Eunice Kennedy Shriver National Institute of Child Health and Human Development of the National Institutes of Health under the Awards R01-HD097331/HD/NICHD (PEF), the National Institute of Deafness and Other Communication Disorders of the National Institutes of Health under the Award R01-DC017149 (PEF), the National Institute of Dental and Craniofacial Research under the Award R01DE028576 (MS-C), and the National Institute of Mental Health under the Award R15-MH118692 (DGZ).

# Additional information

### Funding

| Funder | Grant reference number | Author |
| --- | --- | --- |
| Eunice Kennedy Shriver National Institute of Child Health and Human Development | R01-HD097331/HD/NICHD | Paolo Emanuele Forni |
| National Institute on Deafness and Other Communication Disorders | R01-DC017149 | Paolo Emanuele Forni |
| National Institute of Dental and Craniofacial Research | R01DE028576 | Marcos Simoes-Costa |
| National Institute of Mental Health | R15-MH118692 | Damian G Zuloaga |

The funders had no role in study design, data collection and interpretation, or the decision to submit the work for publication.

### Author contributions
Jennifer M Lin, Conceptualization, Data curation, Formal analysis, Supervision, Validation, Investigation, Visualization, Methodology, Writing - original draft, Writing - review and editing; Tyler A Mitchell, Data curation, Formal analysis, Validation, Investigation, Writing - review and editing; Megan Rothstein, Data curation, Formal analysis, Validation, Investigation, Visualization, Writing - review and editing; Alison Pehl, Formal analysis, Investigation; Ed Zandro M Taroc, Raghu R Katreddi, Data curation, Software, Visualization, Writing - review and editing; Katherine E Parra, Formal analysis, Writing - review and editing; Damian G Zuloaga, Marcos Simoes-Costa, Resources, Supervision, Funding acquisition, Project administration, Writing - review and editing; Paolo Emanuele Forni, Conceptualization, Resources, Supervision, Funding acquisition, Methodology, Writing - original draft, Project administration, Writing - review and editing

### Author ORCIDs
Jennifer M Lin http://orcid.org/0000-0002-9197-0816
Tyler A Mitchell http://orcid.org/0000-0002-9647-6024
Marcos Simoes-Costa http://orcid.org/0000-0003-1452-7068
Paolo Emanuele Forni http://orcid.org/0000-0001-6547-3464

## Ethics

All mouse studies were performed according to the approved Institutional Animal Care and Use Committee (IACUC) protocols (#20-002, #19-001) of the University at Albany.

## Decision letter and Author response

Decision letter https://doi.org/10.7554/eLife.77259.sa1
Author response https://doi.org/10.7554/eLife.77259.sa2

# Additional files

## Supplementary files

• Supplementary file 1. Differentially expressed genes in mature apical and basal vomeronasal sensory neurons (VSNs).

• Supplementary file 2. Significantly dysregulated genes when *Tfap2e* is ectopically expressed in apical vomeronasal sensory neurons (VSNs).

• Transparent reporting form

## Data availability

All data generated or analyzed during this study are included in the manuscript and supporting file; Source Data files have been provided for Figures 1 and 6. The scRNA-seq data discussed in this publication have been deposited in NCBI's Gene Expression Omnibus and are accessible through GEO series accession number GSE192746 (https://www.ncbi.nlm.nih.gov/geo/query/acc.cgi?acc=GSE192746). We also utilized previously published data from (Katreddi et al., 2021), available through GEO series accession number GSE190330 (https://www.ncbi.nlm.nih.gov/geo/query/acc.cgi?acc=GSE190330). The data from this CUT&RUN experiment has been deposited into the NCBI's Expression Omnibus and are accessible through GEO series accession number GSE193139 (https://www.ncbi.nlm.nih.gov/geo/query/acc.cgi?acc=GSE193139).

The following datasets were generated:

| Author(s) | Year | Dataset title | Dataset URL | Database and Identifier |
|---|---|---|---|---|
| Lin JM, Taroc EZ, Katreddi RR, Forni PE | 2022 | Single cell RNA sequencing of P10 and adult vomeronasal organ of OMPCreR26AP2e and Controls | https://www.ncbi.nlm.nih.gov/geo/query/acc.cgi?acc=GSE192746 | NCBI Gene Expression Omnibus, GSE192746 |
| Lin JM, Mitchell TA, Rothstein M, Pehl A, Taroc EZ, Katreddi RR, Parra KE, Zuloaga DG, Simoes-Costa M, Forni PE | 2022 | Tfap2e/AP-2ε has both activating and repressive roles in controlling genetic programs of vomeronasal sensory neurons that underlie sociosexual behavior in mice | https://www.ncbi.nlm.nih.gov/geo/query/acc.cgi?acc=GSE193139 | NCBI Gene Expression Omnibus, GSE193139 |

The following previously published dataset was used:

| Author(s) | Year | Dataset title | Dataset URL | Database and Identifier |
|---|---|---|---|---|
| Katreddi R, Lin JM, Taroc EZ, Hicks SM, Forni PE | 2022 | single cell RNA sequencing of adult (P60) vomeronasal organ | https://www.ncbi.nlm.nih.gov/geo/query/acc.cgi?acc=GSE190330 | NCBI Gene Expression Omnibus, GSE190330 |

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
