## [Editor Report]

Lin et al. investigate the role of AP-2e in vomeronasal sensory neurons through targeted gene deletion and rescue. They report that knockout of AP-2e reduces expression of basal markers, while induced expression can rescue basal identity. Moreover, forced expression of AP-2e in mature apical neurons causes them to express some basal markers. Future studies are needed to understand the impact of mutations on VNO receptor expression, and mechanistically why behavioral changes are observed.

---

## [Decision Letter]

**Decision letter after peer review:**

Thank you for submitting your article "Sociosexual behavior requires both activating and repressive roles of Tfap2e/AP-2ε in vomeronasal sensory neurons" for consideration by *eLife*. Your article has been reviewed by 3 peer reviewers, one of whom is a member of our Board of Reviewing Editors, and the evaluation has been overseen by Catherine Dulac as the Senior Editor. The reviewers have opted to remain anonymous.

All reviewers were enthusiastic about the manuscript, but also noted that some additional evidence is required to support key conclusions.

(1) All three reviewers noted that expression of vomeronasal receptors and signaling proteins, like G proteins, are not generally examined throughout the manuscript. Since receptors/G proteins are the key functional expression differences between apical/basal neurons, their expression should be examined in all key experiments, including Figure 1, Figure 5, and Figure 6.

(2) Please pay particular attention to comments 1 and 2 of Reviewer #2 and comment 2 about pS6 data of Reviewer #3.

You will also note that the reviewers have several other comments related to solidifying conclusions in the comments below.

*Reviewer #1 (Recommendations for the authors):*

1) Throughout the manuscript, the authors omit analysis of key functional genes that discriminate apical and basal layers, with the exception of V2R2. In (1) AP-2e knockout mice, (2) basal rescue, and (3) OMP-Cre rescue, it is critical to quantify the number of cells expressing Gi2, Go, and a much larger number of V1Rs, V2Rs, and FPRs. This is needed in almost every figure. (A) How much of the sensory receptor repertoire is governed by AP-2e status? (B) In Figure 1 and Figure 6 (and all other related UMAP analyses), it is critical to show expression of functional apical/basal markers like Gi2 and Go. Are cells really 'reprogrammed' if the receptors and signal transduction elements are not changed?

2) On a related note, Figure 2N indicates a mild 2-fold reduction in V2R2 expression. Why is that sufficient to alter behavior? V2R2 is a broadly expressed V2R – are there other V2Rs that are more substantially impacted?

3) In figure 6, the 'reprogrammed cells' do not appear to be re-clustered in the UMAP plot, as they occupy the same position in UMAP space but are just colored differently (which I assume is the expression of one gene). The apical and basal character instead needs to be defined by the expressed receptor; where are V1Rs and V2Rs in these UMAP plots, and is their expression altered in AP-2e mutants?

4) In Figure 1, VSNs were isolated from OMP-Cre mice. Yet, OMP is only expressed in mature sensory neurons; how is it possible that other early-stage cells were obtained?

5) Are any receptor genes direct targets of AP-2e, and how does it correlate with changes in receptor expression across AP-2e mutants?

6) In Figure 4, the number of cells not the area labeled is the relevant measure.

7) Odor preference is not odor discrimination. The authors make claims about mice distinguishing male/female urine, but the behavioral assay needed for that (an odor dishabituation test) is not used. It would be surprising if mice could not distinguish male/female urine since the MOE can also detect sex-specific volatiles.

8) Sample sizes need to be reported for all experiments- it looks like 4 mice were used for 3D?

9) The manuscript should be extensively edited for language and precision.

10) Figure 6: it is not clear what the difference is between A', A', and A'.

11) Apical vs basal VNO are distinguished by the types of chemicals they detect: V1Rs detect sulfated steroids while FPRs and V2Rs detect peptides. While some V1Rs detect sulfated estrogens, the idea that female odors preferentially activate apical VNOs is rather coarse (Touhara reported a female-specific activator of basal VNOs too, for example in Nature 2005).

*Reviewer #2 (Recommendations for the authors):*

In general, I find the experiments well designed, rigorously executed, and producing results of general interest (meaning beyond olfaction). I think there are a few points they need to clean up:

1. Not surprisingly, considering the timing of the ectopic AP2-epsilon induction, apical VSNs that now adopt a Vmn2R fate, are still projecting to their original territory of the accessory olfactory bulb. This suggests that apical VSNs that respond to inappropriate stimuli activate regions of the bulb that should elicit different responses. In this vein, it is weird that these mice have normal behavioral responses. One explanation would be that the transformed VSNs do not send signals to the brain because the Vmn2R signaling pathway cannot function properly in the transformed neurons. Alternatively, upstream wiring changes may be responsible for the normal behavioral responses. It would be very informative for the understanding of this phenotype for the authors to provide some measure of ligand-induced activity in the OBs of these mice compared to control mice.

2. Although I commend the authors for their efforts to functionally characterize AP2-epsilon binding by CUT&RUN, this analysis is missing important controls. First, CUT&RUN has a tendency to provide false positives, as any accessible region gives some signal regardless of whether the transcription factor is actually bound. The fact that AP2 motifs are the most significantly enriched in this dataset is a good sign for specificity, but it does not really reveal if individual peaks are specific or not. Thus, the authors need to provide control CUT&RUN experiments from the AP2-epsilon KO mice. Further, and rather more importantly, in order to evaluate whether ectopically expressed AP2-epsilon can access binding sites in apical cells that constitute its targets in the basal cells, the authors need a direct comparison between wild type AP2-epsilon CUT&RUN in basal cells vs ectopic AP2-epsilon CUT&RUN in apical cells.

3. The authors should explore the effects of AP2-epsilon expression in olfactory neurons as it is possible that cellular transformation in the main olfactory epithelium contributes to the behavioral phenotype (or the lack of a behavioral change). Also they should show whether V1Rs and their signaling components are still expressed in the transformed VSNs.

*Reviewer #3 (Recommendations for the authors):*

1) The authors have examined the molecular markers of basal and apical VSNn in ectopic AP-2e and control animals using scNRA-seq, IHC, and in situ experiments. However, they did not show the changes of V1Rs and V2Rs that can be achieved by bulk RNA-seq and in situ using specific receptor probes. Although AP-2e may not directly bind to the VR genomic regions as revealed by the CUT & RUN results, expression of VRs can be regulated by other AP-2e targets. This is very critical as the receptors directly detect the sex-specific odorants, which is also related to the following point.

2) I suggest moving Figure 3 to the end and merging with Figure 7 to a single figure or two figures with one having behavioral data and the other having pS6 activation results. Meanwhile, the pS6 activation experiments from the following animals are needed to interpret the behavioral data: (1) male AP-2e KO, AP-2e rescue, and littermate controls to female and male soiled bedding; (2) male ectopic AP-2e and littermate controls to female soiled bedding; (3) female ectopic AP-2e and littermate controls to male soiled bedding. Quantification of the above activation experiments and receptor changes (see point 1) could help to link the molecular data with the behavioral data.

3) The authors stated that ectopic expression of AP-2e in apical VSNs turns on expression of some basa VSN markers. However, the current evidence for this statement is not very convincing. Co-staining of AP-2e, basal VSN markers (Gαo or V2R2 or some V2Rs marked by receptor probe mix), and apical VSN marker (Gai or Meis2 or some V1Rs marked by receptor probe mix) in ectopic AP-2e and littermate controls would elucidate whether the basal VSN marker genes are indeed turned on in apical VSNs.

4) The authors used different words and possibly different genotypes (controls/wild-type/OMP-Cre+/-) in the main text and figures when referring to control animals. Please specify the genotypes for clarity across the whole manuscript and explain why they used different genotypes as controls.

5) The graphical abstract is too complicated and detailed. The authors could do a better job to extract their major findings and depict a more concise one.

[Editors’ note: further revisions were suggested prior to acceptance, as described below.]

Thank you for resubmitting your work entitled "Sociosexual behavior requires both activating and repressive roles of *Tfap2e/AP-2ε* in vomeronasal sensory neurons" for further consideration by *eLife*. Your revised article has been evaluated by Catherine Dulac (Senior Editor) and a Reviewing Editor.

The manuscript has been improved but there are some remaining issues that need to be addressed, as outlined below:

Please revise the text to discuss each point raised below in full. Additional experiments are not requested.

*Reviewer #1 (Recommendations for the authors):*

The authors have performed new experiments in response to the reviewers' comments, but some questions remain.

– In the rebuttal, the authors mention that no changes in V1R expression were observed upon ectopic AP-2e expression. This needs to be mentioned explicitly in the text, whether or not the reason is due to insufficient sequencing data. Similarly, only minimal changes in V2R expression are observed. The extent of reprogramming seems rather subtle across the functional repertoire of the VNO. I am still confused why there are changes in behavior, and changes in apical neuron responsiveness (6E/F) if the vomeronasal receptor repertoire and signaling components are unchanged. This leads me to worry about the quality of behavioral experiments. The key evidence of a behavioral change is in Figure 6G, as 6H provides only negative data. Furthermore, 6G has a p value > 0.05; the phrase 'non-significant reduction' is misleading and should not be used. The authors should provide more data in Figure 7 about the VR repertoire using available data and discuss why behavioral changes might possibly be observed.

– Behavioral experiments in Figure 3 involve a rather small sample size that is identical between B and D; were the same animals re-used for both experiments? More concerning, the knockout phenotype is not VNO-specific and could very well be due to other physiological or neuronal changes in these mice.

---

## [Author Response]

Reviewer #1 (Recommendations for the authors):1) Throughout the manuscript, the authors omit analysis of key functional genes that discriminate apical and basal layers, with the exception of V2R2. In (1) AP-2e knockout mice, (2) basal rescue, and (3) OMP-Cre rescue, it is critical to quantify the number of cells expressing Gi2, Go, and a much larger number of V1Rs, V2Rs, and FPRs. This is needed in almost every figure.

The definition of apical and basal neurons has been historically restricted to what was known. Here by using single cell sequencing we extend the number of molecular markers not limiting the categorization of the two main types of VSNs to what G proteins and receptor subtype they express.

In Figure 2 we immunolabeled and quantified for:

– Meis2 that is a clear and specific marker for neurons with apical identity.

– V2R2 that highlights most of the basal VSNs

– Gao and Gai2 that are classic apical-basal markers

– And we introduced Keratin 18 as a new marker for basal neurons.

The focus of this paper is not the signal transduction of pheromones through specific VR but to understand what grossly defined the identity of the two main types of neurons.

We believe that the markers we showed and quantified in WT, Ap2e Null are sufficient to indicate that reintroduction of Ap-2e rescues the gross morphology of the VNO. The primary scope of this experiment was to validate the functionality of the newly generated R26-AP-2e mouse line.

Regarding the ectopic mice, we have added, throughout ­the manuscript, as further described, new evidence that support our findings (new Figure 4, New figure 7, New Figure 4 —figure supplement 2, Figure 4 —figure supplement 3, Figure 7 —figure supplement 1)

(A) How much of the sensory receptor repertoire is governed by AP-2e status?

None of our experiments indicate a direct role for AP-2e in controlling Vmnr genes. However, we only see significant changes in expression of genes belonging to the Vmn2r C-family (Figure 4 —figure supplement 2). The most obvious expression changes are for Vmn2r7, notably we also found that low mRNA levels for Vmn2r7 can be found in some apical VSNs of controls. Due to the variability in expression among a limited number of analyzed cells we do not know if the expression of other Vmn2r also change.

(B) In Figure 1 and Figure 6 (and all other related UMAP analyses), it is critical to show expression of functional apical/basal markers like Gi2 and Go.

We added Gai2 and Gao in Figure 1. The new version of figure 1 gives a much more complete description of the gene expression of the two main types of VSNs. In old Figure 6 now Figure 7 we included Gai2 and Gao in the heatmap (Figure 7C). Gnao1 is significantly upregulated, but Gai2 has no difference in mRNA expression, reinforcing the idea that, even after ectopic tfap2e, apical VSNs retain the expression of many of their default genes. We now added a new Figure 4 where we visualize the upregulation of Gao in apical cells via RNAscope.

Are cells really 'reprogrammed' if the receptors and signal transduction elements are not changed?

This is a very interesting and philosophical point. We believe that our Sc-Seq data (figure 4, 7, Figure 4 —figure supplement 2, Figure 4 —figure supplement 3, Figure 7 —figure supplement 1) support the conclusion that ectopic expression of tfap2e is sufficient to turn on a considerable number of genes normally enriched the basal cell type transcriptome only and to downregulate apical genes. Therefore ectopic tfap2e changes the transcriptome/program of these cells.

The current idea, mostly driven by Hobert’s lab, is that differentiated neurons cannot be reprogrammed as the chromatin modifications prevent that from happening. Our data indicate that after tfap2e expression we turn on programs that normally are not active in apical neurons, therefore we do, to a certain extent, reprogram the neurons. This reprogramming however does not seem to be sufficient to induce a transdifferentiation. In fact, the apical VSNs that express basal genes still retain the expression of most of their original apical identity genes, including V1Rsignal transduction machinery( Figure 4; Fig6, Figure 4 —figure supplement 2), and guidance receptors (Figure 6 —figure supplement 1) but with the addition of several basal enriched genes (see Figure 7, Supplementary files 1 and 2, Figure 6 —figure supplement 1). These data are in line with newly published observations that suggest that the initiation of the two alternative apical and basal programs is established prior to AP-2e expression (Katreddi et al., 2022).

To better visualize that the cells change their transcriptopme after AP-2e expression, we now added a correlation plot (Figure 7 —figure supplement 1). This shows that while in controls we have sets of genes that correlate in their expression in either apical or basal VSNs, in the mutants, we lose part of the cell type specific correlation.

Please, note that the conclusion of our paper is not that tfap2e turns the apical neurons in basal neurons but that tfap2e is a key player in defining the basal identity and that dysregulation of a TF can alter the transcriptome of differentiated neurons. In this paper we identified several previously unreported basal markers. We see significant changes in those as well as changes in some of the markers with role in controlling signal transduction.

2) On a related note, Figure 2N indicates a mild 2-fold reduction in V2R2 expression. Why is that sufficient to alter behavior?

This is an interesting question that is hard to answer unless we did single cell sequencing of WT and KO, we believe that this could be addressed somewhere else. However, we previously published that tfap2e loss of function leads to a progressive degeneration of the basal VSNs. In figure 2 we used V2R2 as a classic marker to show the phenotype: less V2R cells in the KO, thinner basal layer. The V2R2 antibody broadly recognizes the family C of V2r receptors, and thus gives a broad readout of general V2r expression.

Moreover, we previously showed that without tfap2e some basal VSNs start to express apical genes (Lin et al., 2018). We never stated that the defective behavior of tfap2eKOs (Figure 3) is a result of the sole V2R reduction. As tfap2e controls multiple basal genes it is very likely that the behavioral deficiencies we observe on the KOs, which are clear and significant (Figure 3), might result from mis regulations of multiple genes which translate in a pathological state of the basal neurons.

The point of this experiment, in the narrative of the article, is to shows that reintroducing tfap2e is sufficient to partially restore the tissue homeostasis (FIGURE 2 D,D’,G,J,M) and to reach a functional response to male pheromones (Figure 3). This experiment shows (a) that our R26AP2e mouse works, (b) that exogenous tfap2e is functional. So, whatever is the actual mechanism or mechanisms determining degeneration and reduced response basal VSNs to male stimuli in the KO are rescued.

We show simultaneous changes in multiple markers and a general reduction of the basal population in Tfap2e KOs that, in combination, result in the behavioral defects. We show in the rescue that the number of cells expressing these markers is partially restored and that this is sufficient to rescues the behavioral phenotype.

V2R2 is a broadly expressed V2R – are there other V2Rs that are more substantially impacted?

Our previous RNA seq data from the KOs did not show loss of specific V2R. However, by analyzing our sc-seq data from ectopic mice we noticed that, in these, most of the upregulated V2Rs belong to the family-C.

To show this we now generated a dot plot comparing Vmn2r expression of control apical and ectopic (Figure 4 —figure supplement 2). Interestingly our data suggest that a small percentage of apical VSNs can express low levers of Vmn2r of the C-family. However, we do not have any evidence that these are directly upregulated by tfap2e. Moreover, our data indicate that even though Vmn2r expression is significantly increased in apical neurons after ectopic tfap2e, the cells that express Vmn2r are only a fraction of all the apical VSNs.

The fact that we see upregulation of V2R of the c-family is likely due to the fact that they are represented enough to reach significance. In order to analyze the expression of multiple V2R we would have to largely increase the number of sequenced cells, this is a very expensive experiment that we could develop in a follow-up story focused on this specific aspect.

3) In figure 6, the 'reprogrammed cells' do not appear to be re-clustered in the UMAP plot, as they occupy the same position in UMAP space but are just colored differently (which I assume is the expression of one gene). The apical and basal character instead needs to be defined by the expressed receptor; where are V1Rs and V2Rs in these UMAP plots, and is their expression altered in AP-2e mutants?

In Figure 6, now Figure 7 the UMPAS in A’-B’’’ show the normal and ectopic expression of tfap2e (red) in control and mutants with respect to Meis2 (blue), we increased the size of the font in the figure to make that visible.­ Moreover we made the overlap more visible. Tfap2e and Meis2 are early and reliable markers for the apical basal lineages (see Figure 4 A1,D1 and Figure 5). These two transcription factors are expressed in mutually exclusive way before Ga0 and Gai2 expression fully segregates. We better characterized this in a recently published article (Katreddi R.R. et al., Development 2022)

We generated a new supplementary figure (Figure 4 —figure supplement 2) with volcano plot and a dot plot showing the percentage of cells expressing V2Rs in control apical and mutant apical VSNs (Figure 4 —figure supplement 2). As previously mentioned, most of the ectopically expressed Vmnr2 are of the C family.

…the reprogrammed cells' do not appear to be re-clustered in the UMAP plot

We respectfully disagree with the reviewer. In fact, the apical clusters of the controls and of the tfap2e ectopic do not occupy the same position in the UMAP. To make that more visible we now generated a new version of Figure 6 now Figure 7, (See Figure 7 A’’’’; B’’’’ and A+B’’’’) where we show the UMAP of controls in gray, the UMAP of OMPCreR26AP2e in magenta and an overlay to show segregation of the mature apical populations in the two genotype. As shown in the merged UMAP the apical neurons of controls and mutants do not occupy the same position in the UMAP space.

Notably, the heat map (Figure 7C), which is more informative and detailed than the feature map, shows that the apical neurons of mice with ectopic tfap2e still express most of the apical genes including Gai2 (Supplementary file 2).

4) In Figure 1, VSNs were isolated from OMP-Cre mice. Yet, OMP is only expressed in mature sensory neurons; how is it possible that other early-stage cells were obtained?

We did not use OMPCre to sort and isolate the vomeronasal neuronal cells. As detailed in the Materials and methods the scRNA-Seq was performed using dissociated whole VNOs from OMPCre mice. The feature plot of the VSNs were identified and pulled from a greater population of sequenced cells post-hoc.

5) Are any receptor genes direct targets of AP-2e, and how does it correlate with changes in receptor expression across AP-2e mutants?

The CUT&RUN experiment did not indicate that any direct interaction between Tfap2e and Vmnr genes.

6) In Figure 4, the number of cells not the area labeled is the relevant measure.

The quantification of area was performed on the Gαo ISH. ISH analysis gave us a clear indication of altered expression, but we found it hard to identify individual cells with good confidence. Therefore, we decided to do an unbiassed densitometric analysis, this allowed us to quantify the extension of the areas of Gαo mRNA expression within the VNE. We tried to do cell counts after immunostaining, but we found that the complex patterning of Gαo expression in the epithelium and the variability in immunoreactivity across cells made it difficult to quantify cells in a reliable way. In the new version of the article, we kept the area percentage of tissue reactive for ISH Figure 4G. Please note that as now indicated, in both figure legend and Materials and methods, significance for percentage has been calculated using arcsine transformation values and unpaired two-tailed t test. For V2R2 the quantification we now show number of cells, Figure 4 H. However, we now added new data in Figure 4K-N showing the expression V2R in Meis2 + cells and of Gao in Mei2 + ; Gai2 cells.

7) Odor preference is not odor discrimination. The authors make claims about mice distinguishing male/female urine, but the behavioral assay needed for that (an odor dishabituation test) is not used. It would be surprising if mice could not distinguish male/female urine since the MOE can also detect sex-specific volatiles.

Thanks, we corrected this on the text.

8) Sample sizes need to be reported for all experiments- it looks like 4 mice were used for 3D?

We corrected this

9) The manuscript should be extensively edited for language and precision.

We extensively revised the text.

10) Figure 6: it is not clear what the difference is between A', A', and A'.

We added color legend on the figure and increased the font of the heatmap

This is now figure 7. A’ Tfap2e expression (red); A” Meis2 expression (blue); A”’ Tfap2e and Meis2. While in controls Ap2e is only on basal in the mutant (B1, B’,B”’) it is expressed starting from immature apical neurons. You can see this as an histochemistry in Figure 4A1-D1.

11) Apical vs basal VNO are distinguished by the types of chemicals they detect: V1Rs detect sulfated steroids while FPRs and V2Rs detect peptides. While some V1Rs detect sulfated estrogens, the idea that female odors preferentially activate apical VNOs is rather coarse (Touhara reported a female-specific activator of basal VNOs too, for example in Nature 2005).

We have added the above-mentioned reference. Nonetheless in our hands, as illustrated in the graph (new Figure 6), we do see, regardless of the sex, a clear bias in the activation of apical VSNs after exposure to female soiled bedding. Both tfap2e ectopic males and females showed reduction in apical VSNs activation in response to female urines.

Reviewer #2 (Recommendations for the authors):In general, I find the experiments well designed, rigorously executed, and producing results of general interest (meaning beyond olfaction). I think there are a few points they need to clean up:1. Not surprisingly, considering the timing of the ectopic AP2-epsilon induction, apical VSNs that now adopt a Vmn2R fate, are still projecting to their original territory of the accessory olfactory bulb. This suggests that apical VSNs that respond to inappropriate stimuli activate regions of the bulb that should elicit different responses. In this vein, it is weird that these mice have normal behavioral responses. One explanation would be that the transformed VSNs do not send signals to the brain because the Vmn2R signaling pathway cannot function properly in the transformed neurons. Alternatively, upstream wiring changes may be responsible for the normal behavioral responses. It would be very informative for the understanding of this phenotype for the authors to provide some measure of ligand-induced activity in the OBs of these mice compared to control mice.

By analyzing the gross connectivity of the mice with ectopic AP2-epsilon we did not see obvious changes: apical VSNs projected to the anterior while basal to the posterior AOB (Figure 6 —figure supplement 1). These data suggest that after tfap2e ectopic expression the core of the apical identity is not lost. However, though, in ectopic mice, we found a reduction in the number of activated v1r VSNs in response to female odorants (Fig6C,D,F) we could still find c-Fos activation in the AOB. These data suggest that ectopic tfap2e only partially compromises the functionality of the apical neurons. So, even with ectopic expression of some basal genes, a sufficient number of apical neurons appears to be able to detect, and transduce signals to the brain and to trigger “normal” behaviors in response to stimuli. More details are now provided in new Figure 6.

2. Although I commend the authors for their efforts to functionally characterize AP2-epsilon binding by CUT&RUN, this analysis is missing important controls. First, CUT&RUN has a tendency to provide false positives, as any accessible region gives some signal regardless of whether the transcription factor is actually bound. The fact that AP2 motifs are the most significantly enriched in this dataset is a good sign for specificity, but it does not really reveal if individual peaks are specific or not. Thus, the authors need to provide control CUT&RUN experiments from the AP2-epsilon KO mice. Further, and rather more importantly, in order to evaluate whether ectopically expressed AP2-epsilon can access binding sites in apical cells that constitute its targets in the basal cells, the authors need a direct comparison between wild type AP2-epsilon CUT&RUN in basal cells vs ectopic AP2-epsilon CUT&RUN in apical cells.

In the new version of the manuscript, we added new CUT&RUN data obtained from knockout cells and cells from ectopic expressors Figure 7, and Figure 7 —figure supplement 2. These data indicate that the identified peaks were not false positives. In fact, even though we found some peaks in the TFAP2E KO datasets, these appeared to be largely localized to centromeres/telomeres/repetitive regions. These peaks, that did not overlap with the previously identified peaks in WT animals were now used as 'background' peak.

CUT&RUN from ectopic expressors gave tracks that largely overlap with those of the WT controls (Figure 7; Figure 7 —figure supplement 2). However, plotting the signals at the promoter of apical and basal enriched genes for WT, AP-2e KO and AP-2e ectopic expressors we found more signal at the apical-enriched promoters in the ectopic mice dataset. This data suggests that ectopic expression of AP-2e in apical neurons facilitates its access to the promoter of apical enriched genes as these are normally active in the apical neurons.

3. The authors should explore the effects of AP2-epsilon expression in olfactory neurons as it is possible that cellular transformation in the main olfactory epithelium contributes to the behavioral phenotype (or the lack of a behavioral change). Also they should show whether V1Rs and their signaling components are still expressed in the transformed VSNs.

The single cell seq data indicate that after ectopic Ap-2e there is no significant change in V1Rs expression (at least as mRNAs) between controls and mutants (Figure 4 —figure supplement 2, supplementary file 2). This is the same for Gai2 that however in the VSNs of ectopic animals is, in some cells, co-expressed with Gao. (Figure 4). Based on these observations together with the fact that the mutants’ apical neurons, project to the anterior AOB (Figure 6 —figure supplement 1), and still detect and transduce the signals (Figure 6), we can extrapolate that the apical neurons still express their signaling machinery.

Reviewer #3 (Recommendations for the authors):1) The authors have examined the molecular markers of basal and apical VSNn in ectopic AP-2e and control animals using scNRA-seq, IHC, and in situ experiments. However, they did not show the changes of V1Rs and V2Rs that can be achieved by bulk RNA-seq and in situ using specific receptor probes. Although AP-2e may not directly bind to the VR genomic regions as revealed by the CUT & RUN results, expression of VRs can be regulated by other AP-2e targets. This is very critical as the receptors directly detect the sex-specific odorants, which is also related to the following point.

To address this data, we added supplementary data analyzing the expression of V1R and V2Rs. By analyzing the expression level across apical cells of controls and mutants we do not see significant changes/loss of in V1Rs, suggestive of the fact that AP-2e levels do not (directly or not) change V1Rs expression (Figure 4 —figure supplement 2).

2) I suggest moving Figure 3 to the end and merging with Figure 7 to a single figure or two figures with one having behavioral data and the other having pS6 activation results.

Thank you for the suggestion, we made some changes in the figure organization that we think go well with the narrative.

Meanwhile, the pS6 activation experiments from the following animals are needed to interpret the behavioral data: (1) male AP-2e KO, AP-2e rescue, and littermate controls to female and male soiled bedding; (2) male ectopic AP-2e and littermate controls to female soiled bedding; (3) female ectopic AP-2e and littermate controls to male soiled bedding. Quantification of the above activation experiments and receptor changes (see point 1) could help to link the molecular data with the behavioral data.

1) …..male AP-2e KO, AP-2e rescue, and littermate controls to female and male soiled bedding.

We previously published and shown here that AP-2e KO have a dramatic loss of V2R VSNs, a phenotype similar to that described for Gao KO (Lin et al. 2018). The morphological phenotype is reflected by the loss of aggressive behavior (Figure 3). The morphological changes and the rescue of aggressive behavior in the rescued KO are, in our opinion, very strong evidence of a rescue.

Though this would be a great experiment to add to the paper, we had bad luck with our mouse colony, mice have not been mating successfully and unfortunately, we lost some litters. We did not obtain enough animals to perform the tests requested by the reviewer. To perform these additional observations would now add more than 3 months.

2) ….male ectopic AP-2e and littermate controls to female soiled bedding;

We added these data (Figure 6). The new experiments showed a significant reduction in activation of apical VSNs in response to female bedding in males as observed in females.

3) ….Female ectopic AP-2e and littermate controls to male soiled bedding.

We now added the female data (Figure 6E). We also analyzed the AOBs of control and OMPCre+/-/R26AP-2ε+/- Males and females animals exposed to opposite sex soiled bedding (Figure 6J,K).

This analysis revealed that c-Fos activation in the aAOB was statistically different only after exposure to female soiled bedding (Figure 6I, J). These data suggest that ectopic AP-2ε expression reduces the functionality of the V1R VSNs projecting to the aAOB but does not alter the functionality of basal VSNs.

3) The authors stated that ectopic expression of AP-2e in apical VSNs turns on expression of some basa VSN markers. However, the current evidence for this statement is not very convincing. Co-staining of AP-2e, basal VSN markers (Gαo or V2R2 or some V2Rs marked by receptor probe mix), and apical VSN marker (Gai or Meis2 or some V1Rs marked by receptor probe mix) in ectopic AP-2e and littermate controls would elucidate whether the basal VSN marker genes are indeed turned on in apical VSNs.

We added data in Figure 4, Figure 4 —figure supplement 2 and Figure 4 —figure supplement 3

4) The authors used different words and possibly different genotypes (controls/wild-type/OMP-Cre+/-) in the main text and figures when referring to control animals. Please specify the genotypes for clarity across the whole manuscript and explain why they used different genotypes as controls.

We did not see major difference between OMPCre and WT along the developmental trajectories. For this reason, both have been used as controls depending on availability in the litters of animals of similar sex and age. We specified this in material and methods.

5) The graphical abstract is too complicated and detailed. The authors could do a better job to extract their major findings and depict a more concise one.

We removed graphical abstract, as the journal cannot accommodate it.

[Editors’ note: further revisions were suggested prior to acceptance, as described below.]

Reviewer #1 (Recommendations for the authors):The authors have performed new experiments in response to the reviewers' comments, but some questions remain.– In the rebuttal, the authors mention that no changes in V1R expression were observed upon ectopic AP-2e expression. This needs to be mentioned explicitly in the text, whether or not the reason is due to insufficient sequencing data. Similarly, only minimal changes in V2R expression are observed.

We thank the reviewer for pointing this out. We have updated the discussion with a small new paragraph to explicitly state the limitations of our approach in sensing potential changes in VR genes’ expression (Lines 573-575).

The extent of reprogramming seems rather subtle across the functional repertoire of the VNO. I am still confused why there are changes in behavior, and changes in apical neuron responsiveness (6E/F) if the vomeronasal receptor repertoire and signaling components are unchanged. This leads me to worry about the quality of behavioral experiments. The key evidence of a behavioral change is in Figure 6G, as 6H provides only negative data.

Strictly related to the first point raised by the reviewer, we cannot exclude that subtle changes in the receptor expression levels might occur. Even though we do not have indication of changes in V1R, we did see a reduction in expression levels for ~20% of the apical enriched genes and increased mRNA expression of Gnao1 in apical neurons. Normally Gnao1 is silenced in mature apical cells. If having some ectopic Ga0 can negatively affect signal transduction in apical cells is a possibility that would be nice to explore in the future.

Our data show that our mutant mice have altered expression of calreticulins, these are molecules with roles in vomeronasal receptor transport. So, it could also be possible that ectopic AP-2e could alter processing or membrane presentation of the receptors and therefore signaling. A number of changes can occur at post-transcriptional and post-translational level. At the current state we can only speculate. We have now mentioned this in the manuscript at lines: 586-591.

However, this is the way we interpret the data we obtained and then illustrated in figure 6. Figure 6F shows a reduction in the activity of the apical neurons in response to female urines (Figure 6F). This is a clear indication of an altered responsiveness of the mutants’ apical VSNs to a given stimulus.

Notably, such difference was also reflected by the reduced cFos in aAOB of animals exposed to female urine Figure 6K. The experiment are not just “negative data”.

Notably we noticed more subtle reduction in the responsiveness of apical neurons also to male urines, however, in this case the change resulted to be non-significant.

The fact that the observed differences in functionality do not translate in significant behavioral changes in opposite sex presences for us is an indication that the animals retain sufficient signaling to reach a given threshold that leads to a “normal’ response.

Why there is a reduction in responsiveness in the apical VSNs to female urines we do not know.

Furthermore, 6G has a p value > 0.05; the phrase 'non-significant reduction' is misleading and should not be used.

We changed lines 326-328 to:

“By performing resident intruder tests, we showed that the level of intermale aggression of OmpCre/R26AP-2ε male mice was higher but not significantly different than controls (P=0.051) (Figure 6G)” and lines 350-352 to: “C-Fos activation in the aAOB of female Omp^Cre^/R26AP-2ε mice exposed to male bedding was on average lower than, but not significantly different from controls (Figure 6J).”

The authors should provide more data in Figure 7 about the VR repertoire using available data and discuss why behavioral changes might possibly be observed.

As mentioned in the text about the data in Figure 7 we did not have any evidence that AP-2e directly controls V2R expression. So including data about the V2Rs there would be misleading.

However, data illustrating changes in VR are part of Figure 4 supplement 2. This is as far as we could go with our available data.

– Behavioral experiments in Figure 3 involve a rather small sample size that is identical between B and D; were the same animals re-used for both experiments?

Yes, the same animals were used for both tests. A statement explicitly stating this has been added to the manuscript at lines: 768-769. This was done following our IACUC guidelines to reduce number of animals whenever is possible.

Animals were first tested for opposite sex preference then for aggressive behavior.

Please note that for aggressive behavior there is no need for the animals to be naïve. In fact, as explained in the methods test subjects were housed with intact females for at least one week prior to testing. On the day of testing, all subjects (residents and intruders) were acclimated to the experimental environment for at least 30 minutes prior to the assay. Females were removed immediately before testing.

More concerning, the knockout phenotype is not VNO-specific and could very well be due to other physiological or neuronal changes in these mice.

The statement is correct, the AP2eKO is a full KO. However, we previously explored Cre lineage tracing showing very limited AP2e expression to the VNO and subsets of mitral cells. Moreover, we have previously published that without AP2e the basal VSNs degenerate in a similar fashion to what reported for Gao, Ggamma8, TRPC2 KOs. So, vit a progressive loss of V2R+ neurons a reduction in basal VSNs dependent behavior (e.g. aggressive behavior), was expected. If the behavioral changes that we observed are solely dependent on AP2e LOF in the VNO is a point that we cannot resolve here.

However, we observed that re-expressing AP2e using our new R26AP2e mouse line was sufficient to (a) restore the basal VSNs (b) rescue the aggressive behavior. This experiment was mostly intended to validate our R26 model and the biological functionality of the R26Ap2e model.